# Name Your Colour For the Task: Artificially Discover Colour Naming via Colour Quantisation Transformer

## Abstract

The long-standing theory that a colour-naming system evolves under the dual pressure of efficient communication and perceptual mechanism is supported by more and more linguistic studies including the analysis of four decades' diachronic data from the Nafaanra language. This inspires us to explore whether artificial intelligence could evolve and discover a similar colour-naming system via optimising the communication efficiency represented by high-level recognition performance. Here, we propose a novel colour quantisation transformer, CQFormer, that quantises colour space while maintaining the accuracy of machine recognition on the quantised images. Given an RGB image, Annotation Branch maps it into an index map before generating the quantised image with a colour palette, meanwhile the Palette Branch utilises a key-point detection way to find proper colours in palette among whole colour space. By interacting with colour annotation, CQFormer is able to balance both the machine vision accuracy and colour perceptual structure such as distinct and stable colour distribution for discovered colour system. Very interestingly, we even observe the consistent evolution pattern between our artificial colour system and basic colour terms across human languages. Besides, our colour quantisation method also offers an efficient quantisation method that effectively compresses the image storage while maintaining a high performance in high-level recognition tasks such as classification and detection. Extensive experiments demonstrate the superior performance of our method with extremely low bit-rate colours. We will release the source code upon acceptance.

## 1 Introduction

*Hath not a Jew eyes?*
*Hath not a Jew hands, organs, dimensions, senses, affections, passions?*

William Shakespeare "The Merchant of Venice"

Does artificial intelligence share the same perceptual mechanism for colours as human beings? We aim to explore this intriguing problem through AI simulation in this paper.

Colour involves the visual reception and neural registering of light stimulants when the spectrum of light interacts with cone cells in the eyes. Physical specifications of colour also include the reflective properties of the physical objects, geometry incident illumination, *etc.* By defining a colour space (Forsyth & Ponce, 2002), people could identify colours directly according to these quantifiable coordinates.

Compared to the pure physiological nature of hue categorisation, the complex phenomenon of colour naming or colour categorisation spans multiple disciplines from cognitive science to anthropology. Solid diachronic research (Berlin & Kay, 1969) also suggests that human languages are constantly evolving to acquire new colour names, resulting in an increasingly fine-grained colour naming system. This evolutionary process is hypothesised to be under the pressure of both communication efficiency and perceptual structure. Communication efficiency requires shared colour partitioning to be communicated accurately with a lexicon as simple and economical as possible. Colour perceptual structure is relevant to human perception in nature. For example, the colour space distance

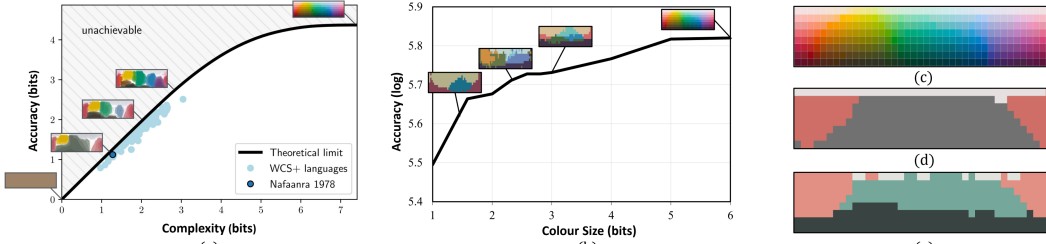

Figure 1: (a) the theoretical limit of efficiency for colour naming (black curve) and cases of the WCS probability map of human colour language copied from Zaslavsky et al. (2022). (b) the colour size (from 1-bit to 6-bit)-accuracy curve on the tiny-imagenet-200 (Le & Yang, 2015) dataset. The WCS probability maps generated by our CQFormer are also shown along the curve. (c) the colour naming stimulus grid used in the WCS (Kay et al., 2009). (d) the three-term WCS probability map of CQFormer after embedding 1978 Nafaanra three-colour system ((light ('fiNge'), dark ('wOO'), and warm or red-like ('nyiE')) into the latent representation. (e) the four-term WCS probability map of CQFormer evolved from (d). The evolved fourth colour, yellow-green, is consistent with the prediction of basic colour term theory (Berlin & Kay, 1969)

between nearby colours should correspond to their perceptual dissimilarity. This structure of perceptual colour space has long been used to explain colour naming patterns across languages. A recent analysis of human colour naming systems, especially in Nafaara, contributes the very direct evidence to support this hypothesis through the employment of Shannon's communication model (Shannon, 1948). Very interestingly, this echos the research on colour quantisation, which quantises colour space to reduce the number of distinct colours in an image.

Traditional colour quantisation methods (Heckbert, 1982; Gervautz & Purgathofer, 1988; Floyd & Steinberg, 1976) are *perception-centred* and generate a new image that is as visual perceptually similar as possible to the original image. These methods group similar colours in the colour space and represent each group with a new colour, thus naturally preserving the perceptual structure. Instead of prioritising the perceptual quality, Hou *et al.* (Hou et al., 2020) proposed a *task-centred/machine-centred* colour quantisation method, ColorCNN, focusing on maintaining network classification accuracy in the restricted colour spaces. While achieving an impressive classification accuracy on even a few-bit image, ColorCNN only identifies and preserves *machine-centred* structure, without directly clustering similar colours in the colour space. Therefore, this pure *machine-centred* strategy sacrifices perceptual structure and often associates nearby colours with different quantised indices.

Zaslavsky *et al.* (Zaslavsky et al., 2022) measure the communication efficiency in colour naming by analysing the informational complexity based on the information bottleneck (IB) principle. Here, we argue that the network recognition accuracy also reflects the communication efficiency when the number of colours is restricted. Since the human colour naming is shaped by both perception structure and communication efficiency (Zaslavsky et al., 2019a), we integrate the need for both *perception* and *machine* to propose a novel end-to-end colour quantisation transformer, CQFormer, to discover the artificial colour naming systems.

As illustrated in Fig. 1.(b), the recognition accuracy increases with the number of colours in our discovered colour naming system. Surprisingly, the complexity-accuracy trade-offs are similar to the numerical results (Fig. 1.(a)) independently derived from linguistic research (Zaslavsky et al., 2022). What is more, after embedding 1978 Nafaanra three-colour system (Nafaanra-1978, Fig. 1.(d)) into the latent representation of CQFormer, our method automatically evolves the fourth colour closed to yellow-green, matching the basic colour terms theory (Berlin & Kay, 1969) summarised in different languages. Berlin and Kay found universal restrictions on colour naming across cultures and claimed languages acquire new basic colour category in a strict chronological sequence. For example, if a culture has three colours (light ('fiNge'), dark ('wOO'), and warm or red-like ('nyiE') in Nafaanra), the fourth colour it evolves should be yellow or green, exactly the one (Fig. 1.(e)) discovered by our CQFormer.

The pipeline of CQFormer, shown in Fig. 2, comprises two branches: Annotation Branch and Palette Branch. Annotation Branch annotates each pixel of the input RGB image with the proper quantised colour index before painting the index map with the corresponding colour in the colour palette. We localise the colour palette in the whole colour space with a novel Palette Branch which detects the

key-point with explicit attention queries of transformer. During the training stage, as illustrated in the red line and black line of Fig. 2, Palette Branch interacts with an input image and reference palette queries to maintain the perceptual structure by reducing the perceptual structure loss. This *perception-centred* design groups similar colours and ensures the colour palette sufficiently represents the colour naming system defined by the World Color Survey (WCS) colour naming stimulus grids. As shown in Fig. 2.(b), each item in colour palette (noted by an asteroid) lies in the middle of corresponding colour distribution in the WCS colour naming probability map. Finally, the quantised image is passed to a high-level recognition module for *machine accuracy* tasks such as classification and detection. Through the joint optimisation of CQFormer and consequent high level module, we are able to balance both *perception* and *machine*. Apart from automatically discovering the colour naming system, our CQFormer also offers an effective solution to extremely compress the image storage while maintaining a high performance in high-level recognition tasks.

Our contributions could be summarised as following:

- We propose a novel end-to-end colour quantisation transformer, CQFormer, to artificially discover colour naming system by considering the needs of both perception and machine. The discovered colour naming system shows similar pattern as human language on colour.
- We propose a novel colour palette generation method that takes the colour quantisation as an attention-based key-point detection task, inputs independent attention queries to generate 3D coordinates, and then automatically finds the selected colour in the whole colour space.
- Our colour quantisation achieves superior performance on both classification and object detection with extremely low bit-rate colours. The source code will be released to facilitate the community.

## 2 RELATED WORKS

**Colour Quantisation Methods:** Traditional colour quantisation (Orchard & Bouman, 1991; Deng et al., 1999; Liang et al., 2003; Wu, 1992) reduces the colour space while maintaining visual fidelity. These *perception-centred* methods usually cluster colours to create a new image as visually perceptually similar as possible to the original image. For example, MedianCut (Heckbert, 1982) and OCTree, (Gervautz & Purgathofer, 1988) are two representative clustering-based algorithms. Dithering (Floyd & Steinberg, 1975) eliminates visual artefacts by including a noise pattern. The colour-quantised images can be expressed as indexed colour (Poynton, 2012), and encoded with PNG (Boutell, 1997). Recently, Hou *et al.* propose a pure *machine-centred* CNN-based colour quantisation method, ColorCNN (Hou et al., 2020), which effectively maintains the informative structures under an extremely constrained colour space. In addition to colour quantisation, Camposeco *et al.* also design a task-centred image compression method for localisation in 3D map data (Camposeco et al., 2019). Since human colour naming reflects both perceptual structure and communicative need, our CQFormer also considers both *perception* and *machine* to artificially discover the colour naming system.

**World Color Survey:** The World Color Survey (WCS) (Kay et al., 2009) comprises colour name data from 110 languages of non-industrialised societies (Zaslavsky et al., 2019b), with respect to the stimulus grid shown in Fig. 1(c). There are 320 colour chips in colour naming stimulus grids, and each chip is at its maximum saturation for that hue/lightness combination, while columns correspond to equally spaced Munsell hues and rows to equally spaced Munsell values. Participants were asked to name the colour of each chip to record the colour naming system.

## 3 METHODOLOGY

### 3.1 PROBLEM FORMULATION

For an image-label pair $(\boldsymbol{x}, y)$ in dataset $\mathcal{D}$, the recognition network $f_\theta(\cdot)$ encodes input image $\boldsymbol{x}$ to predict its label $\hat{y}$ (*i.e.* class probability in classification task). $f_\theta(\cdot)$ can be optimised by minimising the loss between prediction $\hat{y} = f_\theta(x)$ and ground truth $y$, defined as machine-centred loss $\mathcal{L}_M$, to find the appropriate parameter $\theta^\star$:

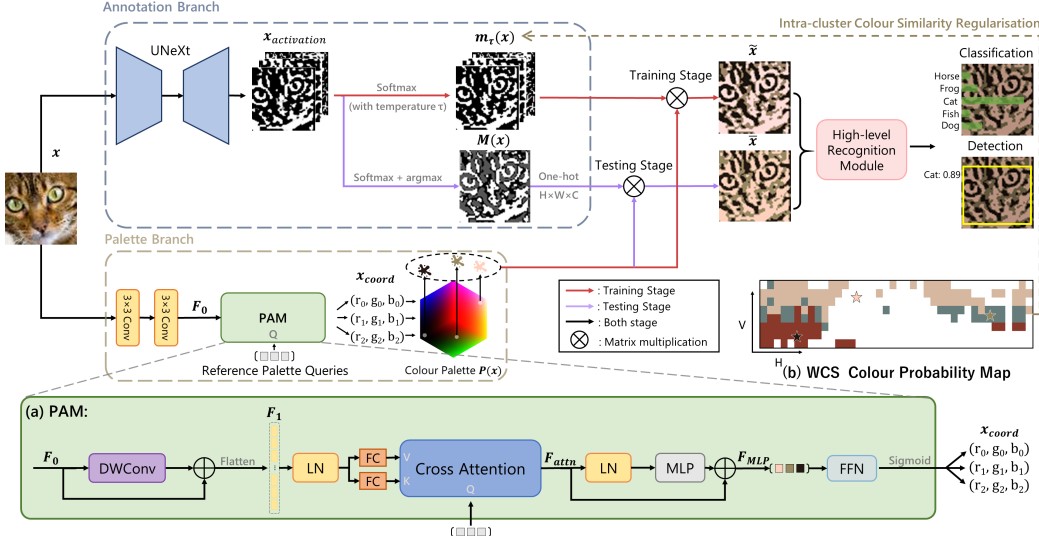

Figure 2: Structure of CQFormer (3-colour quantisation). Red lines are only for training stage; Purple lines are only for testing stage; black lines are both used in both stage. (a) is the detailed structure of PAM. (b) is WCS colour probability map counted across all pixels of the input image.

$$\theta^\star = \arg\min_{\theta} \sum_{(\boldsymbol{x},y)\in D} \mathcal{L}_M\left(y, f_\theta(\boldsymbol{x})\right). \tag{1}$$

We aim to discover the artificial colour naming system by meeting the need for both machine accuracy and human perception. Therefore, CQFormer not only strives for recognition accuracy but also maintains the perceptual structure. The function in Eq. 1 can be rewritten as follow:

$$\psi^\star, \theta^\star = \arg\min_{\psi,\theta} \sum_{(\boldsymbol{x},y)\in D} \mathcal{L}_M\left(y, f_\theta\left(g_\psi(\boldsymbol{x})\right)\right) + \mathcal{L}_P, \tag{2}$$

where $\psi, \theta$ respectively denotes the parameter of CQFormer $g$ and the following high level recognition network $f$. We jointly optimise $g$ and $f$ to find optimal parameters $\psi^\star$ and $\theta^\star$. $\mathcal{L}_P$ is the perceptual structure loss that is perception-centred and will be introduced in detail in Sec. 3.3.

## 3.2 CQFORMER ARCHITECTURE

**Overall Architecture.** An overview of the CQFormer is depicted in Fig. 2, which mainly consists of two branches: (1) Annotation Branch to annotate each pixel of an input RGB image with a proper quantised colour index , and (2) Palette Branch to acquire a suitable colour palette.

Given an input image $\boldsymbol{x} \in \mathbb{R}^{H\times W\times 3}$, Annotation Branch generates a probability map $m_\tau(\boldsymbol{x}) \in \mathbb{R}^{H\times W\times C}$ during training stage or a one-hot colour index map $\text{One-hot}(M(\boldsymbol{x})) \in \mathbb{R}^{H\times W\times C}$ during testing stage. $H$ and $W$ denotes height and width of the input image $\boldsymbol{x}$. $C$ represents the number of quantised colours, and $\tau > 0$ denotes the temperature parameter of the Softmax function (He et al., 2018). In Palette Branch, we first define reference palette queries $\mathbf{Q} \in \mathbb{R}^{C\times d}$ consisting of $C$ learnable vectors of dimension $d$. Each vector represents an automatically mined colour. The $\mathbf{Q}$ interacts with the keys $\mathbf{K} \in \mathbb{R}^{(\frac{HW}{16})\times d}$ and values $\mathbf{V} \in \mathbb{R}^{(\frac{HW}{16})\times d}$ generated from the input image $\boldsymbol{x}$ to produce the colour palette $P(\boldsymbol{x}) \in \mathbb{R}^{C\times 3}$ that consists of $C$ triples of $(R, G, B)$. Each triple represents one of the machine-discovered $C$ colours. Finally, CQFormer outputs the quantised image by calculating a matrix multiplication between $m_\tau(\boldsymbol{x})$ and $P(\boldsymbol{x})$ during the training stage. For testing stage, we get it from $\text{One-hot}(M(\boldsymbol{x}))$ and $P(\boldsymbol{x})$. Then, we feed the colour-quantised image into the high-level recognition module for recognition tasks, *i.e.*, classification and object detection.

**Annotation Branch.** The first component of Annotation Branch is a UNeXt (Valanarasu & Patel, 2022) encoder that outputs per-pixel categories. Given the input image $\boldsymbol{x}$, the encoder generates a class activation map $\boldsymbol{x}_{activation} \in \mathbb{R}^{H\times W\times C}$ with crucial and semantically rich features.

*(1) Testing stage:* We then use the class activation map $\boldsymbol{x}_{activation}$ as the input to a Softmax function over $C$ channels followed by a $\arg\max$ function to obtain a colour index map $M(\boldsymbol{x}) \in \mathbb{R}^{H\times W}$:

$$M(\boldsymbol{x}) = \arg\max_{C}(\text{Softmax}(\boldsymbol{x}_{activation})). \tag{3}$$

After that, a one-hot colour index map $\text{One-hot}(M(\boldsymbol{x})) \in \mathbb{R}^{H \times W \times C}$ is generated from the $M(\boldsymbol{x})$ using a one-hot function $\text{One-hot}(\cdot)$ and finally we utilise the $\text{One-hot}(M(\boldsymbol{x}))$ and the colour palette $P(\boldsymbol{x})$ to generate the test-time colour-quantised image $\bar{\boldsymbol{x}}$ via a matrix multiplication:

$$\bar{\boldsymbol{x}} = \text{One-hot}(M(\boldsymbol{x})) \otimes P(\boldsymbol{x}), \tag{4}$$

where $\otimes$ represents matrix multiplication.

*(2) Training stage:* Since the $\arg\max$ function is not differentiable, we utilise the $\text{Softmax}$ function instead of the $\arg\max$. To avoid over-fitting, we add a temperature parameter $\tau$ (He et al., 2018) to the $\text{Softmax}$ function, forcing the probability map's distribution closer to a one-hot vector, and obtain the probability map $m_\tau(\boldsymbol{x})$ with the temperature parameter $\tau$. The $m_\tau(\boldsymbol{x})$ is computed as:

$$m_\tau(\boldsymbol{x}) = \text{Softmax}(\frac{\boldsymbol{x}_{activation}}{\tau}). \tag{5}$$

As extensively discussed in He et al. (2018), when in a low temperature ($0 < \tau < 1$), the output is similar to a one-hot vector with large diversity; while in a high temperature ($\tau > 1$), the output distribution is similar to a uniform distribution with small diversity. Therefore, we fix the $0 < \tau < 1$ to approximate the $\text{One-hot}(M(\boldsymbol{x}))$ using $m_\tau(\boldsymbol{x})$ during the training procedure. The train-time colour-quantised image $\widetilde{\boldsymbol{x}}$ is generated as:

$$\widetilde{\boldsymbol{x}} = m_\tau(\boldsymbol{x}) \otimes P(\boldsymbol{x}). \tag{6}$$

**Palette Branch.** We locate the representative colours by an attention-based key-point detection strategy, which is originally designed to utilise transformer queries to automatically find key-point location (*i.e.* bounding-box (Carion et al., 2020), human pose (Li et al., 2021)) by attention mechanism. In other words, we reformulate the problem of colour quantisation as a 3D spatial key-point localisation task among the whole RGB colour space. Given the input image $\boldsymbol{x}$, we first extract a high-dimensional and lower resolution feature $F_0 \in \mathbb{R}^{\frac{H}{4} \times \frac{W}{4} \times d}$ by two stacked convolutional layers. After that, the $F_0$ is fed into a palette acquisition module (PAM) to acquire the colour palette $P(\boldsymbol{x})$. As shown in Fig. 2(a), we use a depth-wise convolution (DWConv) (Chu et al., 2021) with a residual connection to complete position encoding, which is suitable for different input resolutions. After that, since the cross attention block expects sequence as input, we collapse the spatial dimensions into one dimension and obtain the flattened feature map $F_1 \in \mathbb{R}^{(\frac{HW}{16}) \times d}$:

$$F_1 = \text{Flatten}(\text{DWConv}(F_0) + F_0). \tag{7}$$

Different from DETR (Carion et al., 2020), where $\mathbf{Q}$, $\mathbf{K}$ and $\mathbf{V}$ are generated from the same input, our $\mathbf{Q}$ are explicit learnable embeddings, called reference palette queries, representing each automatically mined colour. $\mathbf{K}$ and $\mathbf{V}$ are the features extracted from $F_1$ via a Layernorm (LN), which is followed by two fully connected layers (FCs). Generally, we have $\mathbf{Q} \in \mathbb{R}^{C \times d}$, $\mathbf{K}, \mathbf{V} \in \mathbb{R}^{(\frac{HW}{16}) \times d}$. The attention matrix $F_{attn} \in \mathbb{R}^{C \times d}$ is computed by the cross-attention (CA) mechanism as

$$F_{attn} = Attention(\mathbf{Q}, \mathbf{K}, \mathbf{V}) = \text{Softmax}(\mathbf{Q}\mathbf{K}^\top / \sqrt{d})\mathbf{V}. \tag{8}$$

Next, a multi-layer perceptron (MLP) that has two FCs with GELU non-linearity between them is utilised for further feature transformations. The LN is added before MLP, and the residual connection is employed only for MLP. The whole process is formulated as

$$F_{MLP} = \text{MLP}(\text{LN}(F_{attn})) + F_{attn}, \tag{9}$$

where the $F_{MLP} \in \mathbb{R}^{C \times d}$ is the MLP features. Finally, the $F_{MLP}$ would pass through a feed forward network (FFN), which has the same architecture as MLP, followed by a Sigmoid function to get the final $C$ sets of 3D coordinates $\boldsymbol{x}_{coord} \in [0, 1]^{C \times 3}$, which corresponds to values from 0 to 255 in the whole RGB colour space. Hence, we take $255 \times$ of the $\boldsymbol{x}_{coord}$ as the colour palette $P(\boldsymbol{x})$:

$$P(\boldsymbol{x}) = 255 \times \text{Sigmoid}(\text{FFN}(F_{MLP})). \tag{10}$$

**High-Level Recognition Module.** High-level recognition module takes the generated $\bar{\boldsymbol{x}}$ or $\widetilde{\boldsymbol{x}}$ as inputs, and we adopt the popular Resnet (He et al., 2016) as the classifier and the Faster-RCNN (Ren et al., 2015) as the detector.

### 3.3 PERCEPTUAL STRUCTURE LOSS

The CQFormer is trained in an end-to-end fashion using a machine-centred loss $\mathcal{L}_M$. To balance the machine accuracy and perceptual structure, we also add a perceptual structure loss $\mathcal{L}_P$ including perceptual similarity loss $\mathcal{L}_{\text{Perceptual}}$, diversity regularisation $R_{\text{Diversity}}$ and intra-cluster colour similarity regularisation $R_{\text{Colour}}$. The perceptual structure loss can be summarised as follows:

$$\mathcal{L}_P = \alpha R_{\text{Colour}} + \beta R_{\text{Diversity}} + \gamma \mathcal{L}_{\text{Perceptual}}, \tag{11}$$

where $\alpha$, $\beta$ and $\gamma$ are the weights that control the contributions of $R_{\text{Colour}}$, $R_{\text{Diversity}}$, and $\mathcal{L}_{\text{Perceptual}}$, respectively. We combine the $\mathcal{L}_M$ and the $\mathcal{L}_P$ as the total loss $\mathcal{L}$, and then minimise $\mathcal{L}$ for the clssification and detection task:

$$\mathcal{L} = \mathcal{L}_M + \mathcal{L}_P. \tag{12}$$

Additionally, we define WCS colour probability map $m_{\text{WCS}} \in \mathbb{R}^{8 \times 40 \times C}$ on the colour naming stimulus grids in the WCS and use it to calculate $R_{\text{Colour}}$. As shown in Fig. 2.(b), the horizontal axis represents the Munsell hue ranging from $0°$ to $360°$, and the vertical axis denotes the Munsell value ranging from 0 to 1. Each grid is associated with the equally spaced (hue, value) coordinate and a grid's colour index $c$ with the $C$ corresponding occurrence frequencies. The map for human language is collected by the participants' colour perception in WCS (see in Fig. 3(c)), and the machine one is counted across the pixels' colour index in the datasets (see in Fig. 1(d), (e) ,Fig. 2.(b)). Precisely, we first map the colour index of each pixel to the grid with the same (hue, value) coordinate as the pixel and then count the frequency of occurrence of each colour index in each grid. Finally, we take the colour index with the most considerable frequency as the grid's colour index.

**Perceptual Similarity Loss:** The perceptual similarity loss $\mathcal{L}_{\text{Perceptual}}$ is a mean squared error (MSE) loss between the quantised image $\widetilde{\boldsymbol{x}}$ and the input image $\boldsymbol{x}$, which makes the $\widetilde{\boldsymbol{x}}$ or $\bar{\boldsymbol{x}}$ perceptually similar to the $\boldsymbol{x}$. In particular, the $\mathcal{L}_{\text{Perceptual}}$ ensures each item of the $P(\boldsymbol{x})$ (noted by asteroids in Fig. 2(b)) lies in the centre of corresponding colour distribution in the WCS colour naming probability map. The $\mathcal{L}_{\text{Perceptual}}$ is formulated as:

$$\mathcal{L}_{\text{Perceptual}} = \mathcal{L}_{\text{MSE}}(\widetilde{\boldsymbol{x}}, \boldsymbol{x}), \tag{13}$$

**Diversity Regularisation:** To encourage the CQFormer to select at least one pixel of all $C$ colours, we adopt the diversity regularisation term $R_{\text{Diversity}}$ proposed by Hou et al.. Diversity is a simple yet efficient metric that serves as an unsupervised signal to maintain colour diversity in the quantised image by maximising the maximum probability of each channel. The $R_{\text{Diversity}}$ is calculated as:

$$R_{\text{Diversity}} = \log_2 C \times \left( 1 - \frac{1}{C} \times \sum_c \max_{(u,v)} [m_\tau(\boldsymbol{x})]_{u,v} \right) \tag{14}$$

**Intra-cluster Colour Similarity Regularisation:** The CQFormer associates each pixel of the $\boldsymbol{x}$ with a colour index, and the pixels with the index $c$ form a cluster $Cluster_c$ covering a part of the $m_{\text{WCS}}$ (see the different colour grids in Fig. 2.(b)). To make sure the pixels in the same cluster are perceptually similar in colour as possible, we propose the intra-cluster colour similarity regularisation $R_{\text{Colour}}$ and calculate it in the WCS's Munsell HSV colour space. At first, we calculate the centroid colour value $\mu_c$ of $Cluster_c$. Then, we compute the squared colour distance $\text{Dist}^2_{\text{HSV}}$ between all pixels in $Cluster_c$ and $\mu_c$ in the conical representation of HSV. Finally, we calculate the mean value of $\text{Dist}^2_{\text{HSV}}$ for each cluster and obtain the $R_{\text{Colour}}$:

$$\text{Dist}^2_{\text{HSV}}\{[h_1, s_1, v_1], [h_2, s_2, v_2]\} = (v_2 - v_1)^2 + s_1^2 v_1^2 + s_2^2 v_2^2 - 2 s_1 s_2 v_1 v_2 \cos(h_2 - h_1). \tag{15}$$

$$R_{\text{Colour}} = \frac{1}{C} \times \sum_c \frac{1}{N_c} \sum_{i=1}^{N_c} \text{Dist}^2_{\text{HSV}}\{\boldsymbol{x}_c[i], \mu_c\}, \tag{16}$$

where $N_c$ is the number of all pixels in $Cluster_c$, and $\boldsymbol{x}_c[i]$ represents the Munsell HSV value of the i-th pixel in $Cluster_c$.

## 3.4 COLOUR EVOLUTION

To discover the colour naming system, we explore the colour evolution based on the classification task using the CQFormer cascaded with the Resnet-18 (He et al., 2016) without any pre-train models. It consists of two successive stages. Since there are various colour naming systems associated with corresponding languages, the first embedding stage aims to embed the colour perceptual knowledge of a certain language into the latent representation of the CQFormer. For example, CQFormer first learns and matches the Nafaanra with three colours. Specifically, we design two embedding solutions to either distil full colour probability map embedding or only representative colours to CQFormer. The second evolution stage then lets CQFormer evolve more colours, *i.e.* splitting the fourth colour from the learned three colour system under the pressure of accuracy without restriction.

**Embedding Stage:** *(1) Colour Probability Map Embedding:* This embedding forces our CQFormer to match the identical WCS colour probability map of a certain language. At first, we train CQFormer to have the same colour number $C$ of the designed human colour system. As illustrated in Fig. 3, for each pixel in the input image, we collect the pixel's spatial position coordinate $(i, j)$ in the input image, and locate its Munsell H and V value at the $m_{\text{WCS}}$ of certain language to find its

corresponding $C$ probability values $p_{\text{Human}}(i, j) \in \mathbb{R}^C$ of human colour categories (*e.g.* 9% for 'fiNge', 4% for 'wOO', and 87% for 'nyiE' in the little red square of Fig. 3(c)). After performing the above operations on all pixels of the input image, a set of probability values is generated $\{p_{\text{Human}}(i, j) \mid i \in [0, H), j \in [0, w)\}$. We arrange each $p_{\text{Human}}(i, j)$ in this set according to its spatial position $(i, j)$ and obtain a new $H \times W \times C$ matrix regarded as human language probability map $m_{\text{Human}}(\boldsymbol{x})$. Then we force the $m_{\tau}(\boldsymbol{x})$ to match the $m_{\text{Human}}(\boldsymbol{x})$ with a cross-entropy loss $\mathcal{L}_{CE}$. Here we utilise the $\arg\max$ on $m_{\text{Human}}(\boldsymbol{x})$ for efficient embedding and replace the loss function in Eq. 12 with Eq. 17 . In this knowledge distillation way, our CQFormer successfully inherits the colour naming system of human language, which is used in the subsequent colour evolution stage.

$$\mathcal{L} = \mathcal{L}_M + \mathcal{L}_{CE}(m_{\tau}(\boldsymbol{x}), \arg\max(m_{\text{Human}}(\boldsymbol{x}))). \tag{17}$$

*(2) Central Colour Embedding:* Alternatively, we could distil less information of human colour naming system. Here, we only embed representative colours and their $C$ central colour H and V coordinates $\mu_{\text{Human},c} \in \mathbb{R}^2, c \in [0, C)$ in human's $m_{\text{WCS}}$ (noted by asteroids in Fig. 3(c)). Specifically, we utilise it to draw the three $m_{\text{WCS}}$s in Fig. 1(b). By optimising the loss function in Eq. 12, where we replace the $\mu_c$ in $R_{\text{Colour}}$ with aforementioned $\mu_{\text{Human},c}$ and ignore the $S$ value of HSV, we obtain the embedded CQFormer with the central colour.

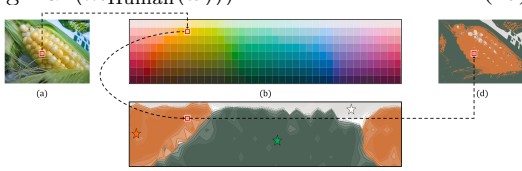

Figure 3: The procedure of colour probability map embedding. (a) the origin RGB image. (b) the WCS colour naming stimulus grids. (c) the $m_{\text{WCS}}$ of Nafaanra-1978. (d) the human colour index map $\arg\max(m_{\text{Human}}(\boldsymbol{x}))$ for Nafaanra-1978.

**Colour Evolution Stage:** After inheriting the existing colour naming system of human language, the CQFormer could evolve to acquire fine-grained new colour on top of the embedded model. In this stage, we remove the restriction of the number of colours $C$ and encourage the CQFormer to evolve more colours under the pressure of accuracy.

## 4 EXPERIMENTS

We evaluate our CQFormer on mainstream benchmark datasets of both image classification (Sec. 4.2) task and object detection task (Sec. 4.3). Additionally, we specifically design a colour evolution experiment (Sec. 4.4) to demonstrate how our CQFormer automatically evolves to increase fine-grained colours. For ablation study, visualisation and detailed results, please refer to Appendix Sec. 6.

### 4.1 DATASETS AND EXPERIMENT SETUP

**Datasets:** For classification, we utilise CIFAR10 (Krizhevsky et al., 2009), CIFAR100 (Krizhevsky et al., 2009), STL10 (Coates et al., 2011) and tiny-imagenet-200 (Tiny200) (Le & Yang, 2015) dataset. Both CIFAR10 and CIFAR100 contain 60000 images, covering 10 and 100 classes of general objects, respectively. The STL10 consists of 13000 images (5000 for training and 8000 for testing) with 10 classes. The Tiny200 is a subset of ImageNet (Deng et al., 2009) and contains 200 categories with 110k images. All train images are random cropped, random horizontal flipped and resized into their respective original resolutions. For object detection, we utilise MS COCO (Lin et al., 2014) dataset, which contains ∼118k images with bounding box annotation in 80 categories. Here, we use COCO `train2017` set as the train set and use COCO `val2017` set as the test set.

**Evaluation Metrics:** For classification, we report top-1 classification accuracy as the evaluation metric. For object detection, we report average precision (AP) value in COCO evaluation setting.

**Implement Details:**

*(i) Upper bound:* We utilise the performance of classifier/detector without additional colour quantisation methods in full colour space (24 bit) as the upper bound. For classification upper bound, we adopt Resnet-18 (He et al., 2016) network. For detection upper bound, we adopt Faster-RCNN (Ren et al., 2015) network with ResNet-50 (He et al., 2016) backbone and FPN (Lin et al., 2017) neck.

*(ii)Training Settings:* All colour quantisation experiments are finished at quantisation levels from 1-bit to 6-bit, *i.e.* $C \in \{2, 4, 8, 16, 32, 64\}$. For classification, we cascade CQFormer with the

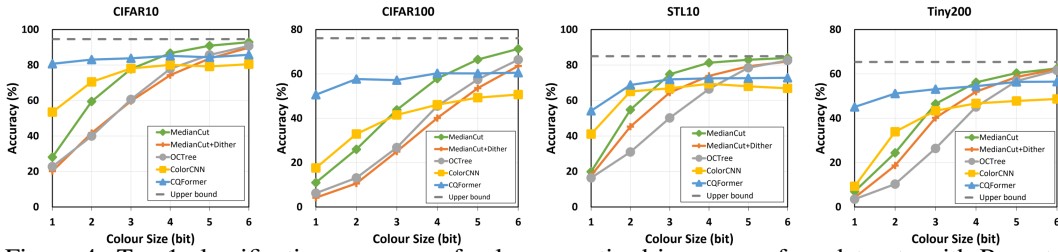

Figure 4: Top-1 classification accuracy of colour-quantised images on four datasets with Resnet-18 (He et al., 2016) networks.

classifier and jointly train them without any pre-trained models on a single GeForce RTX A6000 GPU. We set the temperature parameter $\tau = 0.01$ and the $\alpha = 1$, $\beta = 0.3$ , $\gamma = 1$. We employ an SGD optimiser for 60 epochs using the Cosine-Warm-Restart (Loshchilov & Hutter, 2016) as the learning rate scheduler. A batch size of 128 (STL10 is set to 32), an initial learning rate of 0.05, a momentum is 0.5, a weight decay is 0.001 are used. For detection, only the Resnet-50 backbone of the detector is initialised with Imagenet (Deng et al., 2009) pre-train weights. We also cascade and jointly train the combination for 12 epochs on 4 Tesla V100 GPUs, with the $\alpha = 0$, $\beta = 1$ , $\gamma = 0.1$, and batch size is set to 8. We also adopt the SGD optimiser with an initial learning rate 0.01, momentum 0.9 and weight decay 0.0001, learning rate decay to one-tenth at 8 and 11 epochs.

**Comparison Methods:** As illustrated in Fig. 4, we compare with three traditional perception-centred methods: MedianCut (solid green line) (Heckbert, 1982) , MedianCut+Dither (solid orange line) (Floyd & Steinberg, 1976) and OCTree (solid grey line) (Gervautz & Purgathofer, 1988), and a task-centred CNN-based method ColorCNN (solid yellow line) (Hou et al., 2020). Specifically, for task-centred ColorCNN, we adopt the same training strategy as we did for CQFormer. For the traditional colour quantisation methods, we conduct comparative experiments as described in the ColorCNN (Hou et al., 2020).

Table 1: Object detection results on MS COCO dataset (Lin et al., 2014) with Faster-RCNN Ren et al. (2015) detector, here we report the average precision (AP) value.

| Method | 1-bit | 2-bit | 3-bit | 4-bit | 5-bit | 6-bit | Full Colour (24-bit) |
|---|---|---|---|---|---|---|---|
| Upper bound | - | - | - | - | - | - | 37.2 |
| Median Cut (w/o D) (Heckbert, 1982) | 6.5 | 9.7 | 11.4 | 14.0 | **16.8** | 18.1 | - |
| Median Cut (w D) (Floyd & Steinberg, 1976) | 7.3 | 8.8 | 11.2 | 13.4 | 14.8 | 15.6 | - |
| OCTree (Gervautz & Purgathofer, 1988) | 8.7 | 9.0 | 9.8 | 10.9 | 13.4 | 14.5 | - |
| **CQFormer** | **8.9** | **11.2** | **12.8** | **14.5** | 16.6 | **19.4** | - |

## 4.2 CLASSIFICATION TASK EVALUATION

Fig. 4 presents comparisons to the state-of-the-art methods on the four datasets. Our proposed CQFormer (solid blue line) has a consistent and obvious improvement over all other methods in extremely low-bit colour space (less than 3-bit). Moreover, our CQFormer archives a superior performance than the task-centred method ColorCNN (Hou et al., 2020) under all colour quantisation levels from 1-bit to 6-bit. As the number of the quantised colours increases, the accuracy of the CQFormer continues to improve, which is similar to the complexity-accuracy trade-offs (Fig. 1(a)) independently derived from linguistic research (Zaslavsky et al., 2022).

Similar to the task-centred ColorCNN, we are also inferior to the traditional method under a large colour space (greater than 4-bit), which is an inherent limitation of the CQFormer. As extensively discussed in Hou et al. (2020), this is very normal since traditional methods clusters all pixels to enforce intra-cluster similarity to make a collective decision. Actually, this also explains why our method consistently outperforms pure machine-centred ColorCNN as our method considers the perceptual structure. In addition, we are quite satisfied with the current superior performance on limited colour numbers, as most human languages only use fewer than 3-bit colour terms. This implies that discovering more colours not only compromises the principle of efficiency but also goes contrary to the expectation of better perceptual effects. In other words, the performance of the CQFormer on limited colour categories may hint at the optimal outcome restricted by the unique neurological structure of human vision and cognition, which are, in turn reflected in a wide array of languages.

### 4.3 DETECTION TASK EVALUATION

Table. 1 shows the object detection results on MS COCO dataset Lin et al. (2014) with Faster-RCNN Ren et al. (2015) detector, we report the average precision value (AP) of CQFormer and the comparison methods. We could see that our CQFormer also gains superior performance in object detection task in most colour sizes. We also evaluate CQFormer with more recent SOTA object detectors. For more details, please refer to the Appendix part.

### 4.4 COLOUR EVOLUTION EVALUATION

**Settings:** This experiment is based on the classification task on CIFAR10 (Krizhevsky et al., 2009) dataset using CQFormer and Resnet-18 (He et al., 2016) without any pre-train models. As introduced in Sec. 3.4, in the first embedding stage, we embed the Nafaanra-1978 (Fig. 3(c)) using the colour probability map embedding. Here we set $C = 4$, $\tau = 0.1$ $\alpha = \beta = \gamma = 0$, and force the fourth colour to be split from light ('fiNge'), dark ('wOO'), and warm or red-like ('nyiE'), respectively. Therefore, the number of quantised colours output by CQFormer is still 3, and we optimise the loss function in Eq. 17 for the initial 40 epochs. In the second colour evolution stage, we make the CQFormer inherit the parameters from the first stage and remove the restriction on the last two colour indexes. The $\beta$ is changed as 1, and the loss function in Eq. 12 is minimised for the subsequent 20 epochs.

During the first stage, the WCS colour probability map generated by the embedded CQFormer is shown in Fig.1(d), and similar to Nafaanra-1978 in Fig.3(c). During the second stage, the CQFormer automatically evolves the fourth colour that is split from dark ('wOO') and close to yellow-green (see in Fig. 1(e)), matching the basic colour terms theory (Kay & McDaniel, 1978). However, we are not able to see the fourth colour split from either light ('fiNge') or warm/red-like ('nyiE') in the $m_{WCS}$, since only 3.7% (if split from light ('fiNge')) and 5.9% (if split from warm/red-like ('nyiE')) of all pixels are assigned to the fourth colour, compared with 23.5% (if split from dark ('wOO')). Very interestingly, this phenomenon echoes the evolution of the information bottleneck (IB) colour naming systems (Zaslavsky et al., 2018), where the fourth colour should be spilt from dark in the "dark-light-red" colour scheme. In other words, similar to human language, the colour naming of AI also evolves under the pressure of accuracy and the discovered colour naming system shows similar pattern as human language on colour. (Zaslavsky et al., 2022).

## 5 LIMITATION AND DISCUSSION

While the complexity-accuracy tradeoff of machine-discovered colour terms, as shown in Fig. 1(b), is quite similar to the numerical limit of categorical counterparts for human languages, the current work is still preliminary. As shown in Fig. 1, the newly discovered WCS colour probability map is still quite different from the human one. A more accurate language evolution replication needs to consider more complex variables such as environmental idiosyncrasies, cultural peculiarities, functional necessities, technological maturity, learning experience, and intercultural communication.

Another promising direction would be associating the discovered colours with human colour terms. This would involve much research on Nature Language Processing, and we hope to discuss it with experts from different disciplines in future works. Last but not least, the AI simulation outcome contributes to the long-standing universalist-relativist debate of the linguistic community on colour cognition. Though not entirely excluding the cultural specificities of the colour schemes, the machine finding strongly supports the universalist view that an innate, physiological principle constraints, if not determine, the evolutionary sequence and distributional possibilities of basic colour terms in communities of different cultural traditions. The complexity-efficiency principle is confirmed by the finding that the numerical limitation of colour categories could lead to superior performance on colour-specific tasks, contrary to the intuitive expectation that complexity breeds perfection. The independent AI discovery of the "green-yellow" category on the basis of the fundamental tripartite "dark-light-red" colour scheme points to the congruence of neural algorithms and human cognition and opens a new frontier to test contested hypothesis in the social sciences through machine simulation. We would be more than delighted if this tentative attempt would prove to be a bridge to link scholars of different disciplines for more collaboration and generate more fruitful results.

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

# 6 APPENDIX

## 6.1 ABLATION STUDY

Table 2: Ablation study results under 1-bit colour quantisation, Tiny200 Le & Yang (2015) dataset, Renset18 He et al. (2016) classifier.

| $\tau$ | $R_{\text{Colour}}$ | $R_{\text{Diversity}}$ | $\mathcal{L}_{\text{Perceptual}}$ | Palette Branch | Accuracy |
|---|---|---|---|---|---|
| ✔ | ✔ | ✔ | ✔ | ✔ | **45.1** |
| ✗ | ✔ | ✔ | ✔ | ✔ | 6.9 |
| ✔ | ✗ | ✔ | ✔ | ✔ | 43.9 |
| ✔ | ✔ | ✗ | ✔ | ✔ | 43.7 |
| ✔ | ✔ | ✔ | ✗ | ✔ | 44.6 |
| ✔ | ✗ | ✗ | ✗ | ✔ | 37.3 |
| ✔ | ✔ | ✔ | ✔ | ✗ | 30.4 |

Table 3: Ablation study results of robustness under under different colour quantisation levels from 1-bit to 4-bit, CIFAR10 Krizhevsky et al. (2009) dataset, Renset18 He et al. (2016) classifier.

| | 1 bit | 2 bit | 3 bit | 4 bit |
|---|---|---|---|---|
| CQFormer | **80.7** | **83.1** | **83.8** | **85.2** |
| CQFormer (with colour jitter) | 79.3 | 81.6 | 83.4 | 84.6 |
| CQFormer(with Gaussian blur ) | 80.6 | 81.7 | 81.9 | 83.6 |

As shown in Table 2, we ablate the important elements in our CQFormer , using Tiny200 (Le & Yang, 2015) classification dataset under 1-bit colour quantisation. We investigate the effectiveness of the temperature parameter by setting $\tau = 1.0$, perceptual structure loss by setting any terms of $\alpha$, $\beta$ and $\gamma$ as 0 and Palette Branch by replacing it with a set of centroids in colour space. We also investigate the robustness of CQFormer by adding colour jitter and Gaussian blur using CIFAR10 (Krizhevsky et al., 2009) classification dataset under different colour quantisation levels from 1-bit to 4-bit. Results are shown in Table. 2 and Table. 3

**Influence of temperature parameter:** Without the temperature parameter, a severe accuracy drop (-38.8%) has occurred, which shows that the temperature parameter in our CQFormer can further approximate the One-hot($M(\boldsymbol{x})$) using $m_\tau(\boldsymbol{x})$ during training stage to boost classification accuracy.

**Influence of perceptual structure loss:** With the $R_{\text{Colour}}$, $R_{\text{Diversity}}$ and $\mathcal{L}_{\text{Perceptual}}$, our CQFormer improve the top-1 accuracy by 1.2%, 1.4% and 0.5%, respectively. When we remove all of them, a considerable accuracy drop of -7.8% has occurred. It demonstrates that the perceptual structure loss contributes to better machine accuracy besides maintaining perceptual similarity.

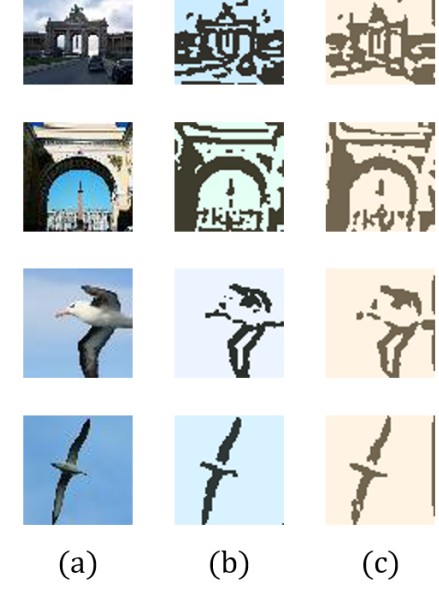

(a)    (b)    (c)

Figure 5: (a) is the original image. (b) is the quantised image with Palette Branch. (c) is the quantised image using a set of centroids instead of Palette Branch.

**Influence of Palette Branch:** If we remove Palette Branch and make the CQFormer just learn a set of centroids in RGB colour space, the accuracy is 30.4%, resulting in a severe drop of 14.7%. As shown in Fig. 5, it would make all the images have the same colour palette rather than the same amount of colours, resulting in a loss of perceptual similarity, *e.g.* the blue sky in Col.(c) is represented as light yellow. Therefore, our Palette Branch ensures that reference palette queries are

sent to interact with the image and creates the colour palette using both machine preference and image perception features, which maintains the colour specificity of each image.

**Robustness of CQFormer:** We add a colour jitter and Gaussian blur to the colour-quantised image. Results are shown in Table. 3. For example, on the CIFAR10 classification dataset, we achieve 79.3%, 81.6%, 83.4%, and 84.6% top-1 accuracy with a colour jitter from 1-bit to 4-bit colour quantisation. The colour jitter causes a little drop of 1.4%, 1.5%, 0.4%, and 0.6%, respectively. Therefore, our CQFormer is robust enough to overcome different noises.

## 6.2 OBJECT DETECTION WITH OTHER DETECTOR

Table 4: Object detection results on MS COCO dataset (Lin et al., 2014) with Sparse-RCNN Sun et al. (2021) detector, here we report the average precision (AP) value.

| Method | 1-bit | 2-bit | 3-bit | 4-bit | 5-bit | 6-bit | Full Colour (24-bit) |
|---|---|---|---|---|---|---|---|
| Upper bound | - | - | - | - | - | - | 45.0 |
| Median Cut (w/o D) (Heckbert, 1982) | 11.5 | 12.7 | 15.4 | 17.0 | 20.4 | 23.2 | - |
| Median Cut (w D) (Floyd & Steinberg, 1976) | 12.3 | 13.8 | 15.2 | 19.6 | 21.8 | 25.6 | - |
| OCTree (Gervautz & Purgathofer, 1988) | 10.7 | 13.4 | 13.2 | 16.7 | 18.9 | 22.8 | - |
| **CQFormer** | **13.9** | **16.5** | **18.8** | **21.5** | **27.5** | **29.8** | - |

To evaluate our CQFormer's generalization on another object detector, we use the recent SOTA object detector Sparse-RCNN Sun et al. (2021) for evaluation. We jointly train our CQFormer with Sparse-RCNN for 36 epochs on 4 Tesla V100 GPUs and adopt AdamW optimizer. The initial learning rate is set to $1.25e^{-6}$, and the learning rate would decay to one-tenth at 24 epochs. The dataset and other training settings are the same as Faster-RCNN experiments in Sec. 4.3.

## 6.3 THE RELATIONSHIP BETWEEN THE IB COLOR NAMING MODEL AND CQFORMER

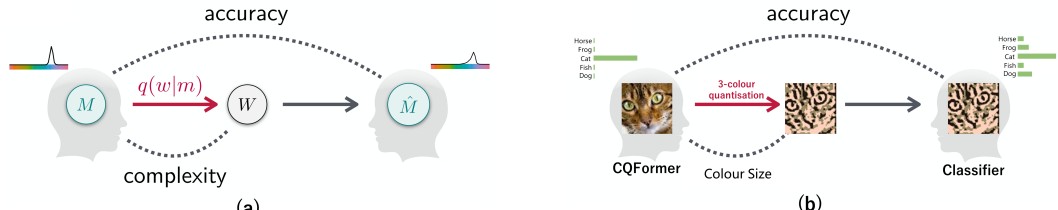

Figure 6: (a) is the IB color naming model proposed by Zaslavsky et al..(b) is our colour quantisation model.

Fig. 6(a) is the information bottleneck (IB) color naming model proposed by Zaslavsky et al.. A straightforward communication scenario, which can be derived from Shannon's communication model (Shannon, 1948), serves as the foundation for this theoretical framework.

Fig. 6(b) is our colour quantisation model. The motivation of our colour quantisation model architecture is driven by the IB color naming model (Zaslavsky et al., 2018). As illustrated in (b), the speaker represents the CQFormer, and the listener represents the classifier. We focus on the case where a colour quantiser (CQFormer) and a classifier communicate about colours. The CQFormer has a "mental" representation, *i.e.* a full-colour image $x$ associated with a prior label $y$ drawn from a prior distribution, and communicates this representation by encoding it into a colour-quantised image $\bar{x}$ according to the perceptual similarity. The classifier receives $\bar{x}$ and attempts to infer from it the full-colour image's label $y$ by predicting the label $\hat{y}$ and constructing another distribution that approximates $y$.

There exists a problem that both the speaker and listener in Fig. 6 (a) have a knowledge of colour naming/recognition at the same time. In contrast, the untrained CQFormer and classifier lack knowledge of colour quantisation and image classification. Therefore, we jointly train both the CQFormer and classifier simultaneously to add prior knowledge of colour quantisation and image classification under a specific bit of colour. Finally, similar to the theoretical limit of semantic efficiency in (Zaslavsky et al., 2018), we obtain optimal image classification accuracy under the specific bit of colour.

## 6.4 Detailed Classification Result

Table. 5 shows the detailed classification result, where the results are the same to Fig. 4.

Table 5: Top-1 classification accuracy of colour-quantised images on four datasets with three networks. We observe that our CQFormer is significantly superior to MedianCut (Heckbert, 1982), OCTree (Gervautz & Purgathofer, 1988), MedianCut+Dither (Floyd & Steinberg, 1976) and Color-CNN (Hou et al., 2020) under low-bit (less than 3-bit) quantisation levels.

| Evaluating Datasets | Methods | Colour size | | | | | | Full Colour (24-bit) |
|---|---|---|---|---|---|---|---|---|
| | | 1-bit | 2-bit | 3-bit | 4-bit | 5-bit | 6-bit | |
| CIFAR10 (Krizhevsky et al., 2009) | baseline | | | | - | | | 94.6 |
| | MedianCut (Heckbert, 1982) | 28.1 | 59.5 | 77.8 | 86.7 | 90.9 | 92.9 | |
| | MedianCut+Dither (Floyd & Steinberg, 1976) | 20.2 | 41.7 | 59.7 | 74.3 | 83.7 | 89.8 | |
| | OCTree (Gervautz & Purgathofer, 1988) | 22.7 | 40.0 | 60.6 | 77.9 | 85.7 | 90.8 | - |
| | ColorCNN (Hou et al., 2020) | 53.5 | 70.5 | 78.1 | 80.1 | 79.2 | 80.5 | |
| | **CQFormer** | 80.7 | 83.1 | 83.8 | 85.2 | 84.4 | 85.8 | |
| CIFAR100 (Krizhevsky et al., 2009) | baseline | | | | - | | | 76.3 |
| | MedianCut (Heckbert, 1982) | 11.0 | 26.0 | 43.8 | 57.9 | 66.5 | 71.3 | |
| | MedianCut+Dither (Floyd & Steinberg, 1976) | 4.2 | 10.6 | 25.0 | 40.1 | 53.6 | 63.6 | |
| | OCTree (Gervautz & Purgathofer, 1988) | 6.2 | 13.1 | 26.8 | 45.5 | 57.5 | 66.4 | - |
| | ColorCNN (Hou et al., 2020) | 17.6 | 32.9 | 41.5 | 46.1 | 49.3 | 50.7 | |
| | **CQFormer** | 50.6 | 57.7 | 57.2 | 60.3 | 60.2 | 60.6 | |
| STL10 (Coates et al., 2011) | baseline | | | | - | | | 84.3 |
| | MedianCut (Heckbert, 1982) | 19.9 | 54.8 | 74.8 | 81.3 | 83.0 | 84.0 | |
| | MedianCut+Dither (Floyd & Steinberg, 1976) | 17.7 | 45.3 | 64.5 | 73.9 | 79.3 | 82.0 | |
| | OCTree (Gervautz & Purgathofer, 1988) | 16.4 | 31.0 | 50.2 | 66.5 | 78.5 | 82.6 | - |
| | ColorCNN (Hou et al., 2020) | 41.1 | 65.1 | 66.9 | 69.45 | 68.0 | 66.9 | |
| | **CQFormer** | 54.2 | 68.8 | 71.9 | 72.5 | 72.6 | 72.8 | |
| Tiny200 (Le & Yang, 2015) | baseline | | | | - | | | 69.1 |
| | MedianCut (Heckbert, 1982) | 7.0 | 24.4 | 46.5 | 56.2 | 60.4 | 62.5 | |
| | MedianCut+Dither (Floyd & Steinberg, 1976) | 4.2 | 18.7 | 40.2 | 52.0 | 58.6 | 62.2 | |
| | OCTree (Gervautz & Purgathofer, 1988) | 3.5 | 10.3 | 26.4 | 45.1 | 56.8 | 61.6 | - |
| | ColorCNN (Hou et al., 2020) | 9.3 | 33.9 | 43.4 | 46.7 | 47.8 | 48.7 | |
| | **CQFormer** | 45.1 | 51.2 | 53.1 | 54.5 | 56.4 | 55.5 | |

## 6.5 Visualisation

As shown in Fig. 7, our CQFormer effectively preserves more perceptual structure and similarity. For instance, aeroplane wings, textures of architecture and vehicle windows.

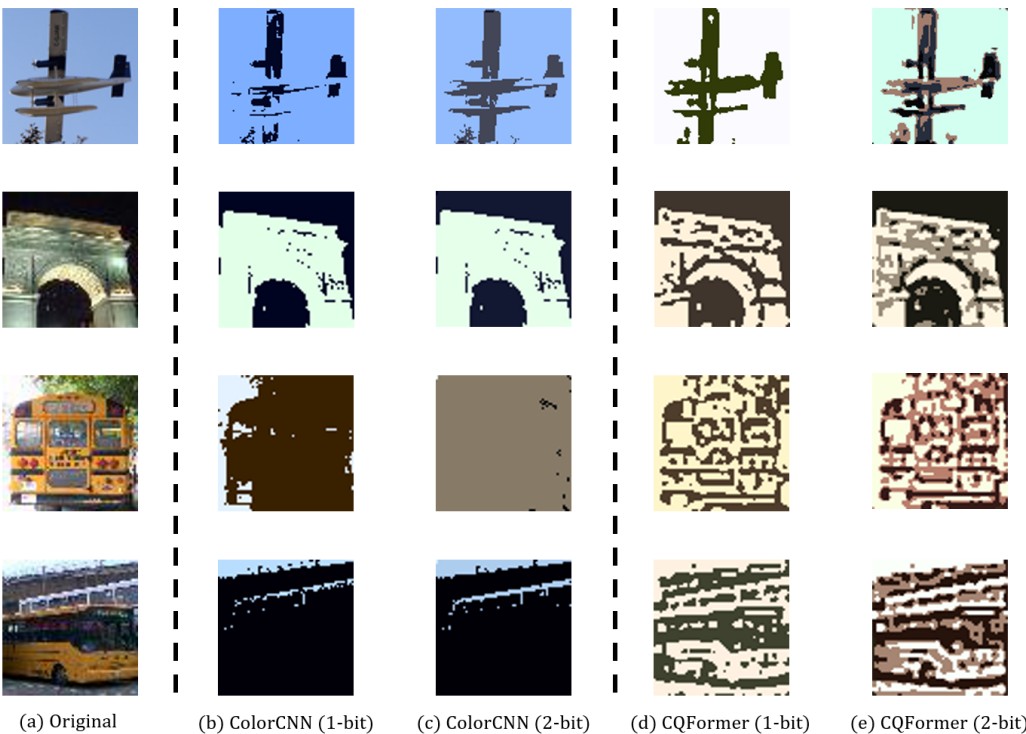

(a) Original    (b) ColorCNN (1-bit)    (c) ColorCNN (2-bit)    (d) CQFormer (1-bit)    (e) CQFormer (2-bit)

Figure 7: Visualisation of 1-bit and 2-bit colour quantisation.

