# OpenReview forum: "Name Your Colour For the Task: Artificially Discover Colour Naming via Colour Quantisation Transformer"
_ICLR.cc/2023/Conference — Submitted to ICLR 2023_

### Official Review · Reviewer_SHap · 2022-10-20

**Confidence:** 4
**Correctness:** 4
**Technical Novelty And Significance:** 2
**Empirical Novelty And Significance:** 2
**Recommendation:** 3

**Clarity, Quality, Novelty And Reproducibility:**

# Clarity
 I found the clarity to vary across sections of the paper. Most of the paper is is clear. but some technical presentations and experiments were hard to follow.
For example:
- Section 3.4 the paragraph "colour probability map embedding" was not clear. Is there any fine-tuning of the model involved here, or just an adaptation of an already trained model? I would suggest a more formal description of the method to clarify the exposition.
- Similarly the paragraph "Colour evolution" in the same section was unclear to me. Please add a more formal description of the approach here.
- The same goes for the last paragraph in section 3.4 "Central colour embedding".  It was not clear to me whether this is an alternative regularisation method used during training, or a post-hoc adaptation of the trained model.

The captions of Figure 4 and Figure 5 are not separated from the main text, probably by applying negative vspace in the latex source, which is hurting readability of the text, and might be in conflict with the author guidelines.

It was not clear to me whether the image classification and detection models are trained jointly with the color quantisation model, or whether the classification/detection models are pre-trained without color quantisation, and the used as is without further training them when learning the colour quantisation model.

The purpose of Section 4.4. is not clear.
For example, the first sentence was unclear to me: "Similar to task centered colorCNN [ref], we are also inferior to the traditional method under a large space."
The section does not describe any experiment, but rather seems to be an interpretation of the previous experiments.
The text itself should be clarified by clearly pointing to the experimental results on which it is based, and integrating it in the corresponding section. In its current state it did not bring me any additional insights.

The colour evolution experiment in section 4.5 it not clearly described. It seem that in the first stage the proposed colour quantisation model is trained to align with the Nafaanra color naming in three colours, and to perform well on the CIFAR-10 classification task. The second stage learns a fourth colour bin, starting from the three colour quantisation. But it seems the design of the experiment already dictates which of the three initial colour bins will be split in the second stage, which seems to strongly direct the outcome of the second training stage.
Moreover, it is not clear if the alternative color quantisation methods would exhibit similar behaviour, or whether this is unique to the proposed method.

# Quality

- The paper refers to studies of color naming in the Nafaanra language, but does not explain why these studies are of particular interest as compared to color naming in other languages, nor how the findings for Nafaanra differ from those for other languages. When referring to analysis of Nafaara in the first paragraph of page 2, or elsewhere, there is no reference to the corresponding study.

- The discussion of related work in section 2 is rather limited, and does not succeed in clearly placing the current work in the context of related work, ie which problems of existing methods are solved? What additional insight does the current work provide over others?

- The technical approach lacks a clear motivation. Why is it insufficient to learn several centroids in color space, and assign pixels to the nearest color centroid? What is the motivation for the Palette and Annotation branches of the proposed approach over simpler alternatives?

- Purity regularisation: it was not clear to me why a max over pixel positions is taken, this seems to ensure that at least one pixel is strongly assigned to each one of the quantisation bins, rather than that *all* pixels are assigned to just one color. Did the authors consider, for example, summing over the pixels the entropy of the assignment distribution?

- When considering image classification, and increasing the number of colour quantisation levels, the proposed method saturates to an accuracy level  that is lower than that for some of the other methods to which is compared (OCTree, MedianCut, MedianCut+dither). A discussion to interpret this result is missing. Is this an inherent limitation of the proposed method? It is due to what? Can it be overcome?


#  Novelty
 The proposed technique to address color quantisation seems novel to me. The topic of study is relatively little studied, and the findings seem to make a contribution to this literature. The technical tools / architectures used in the work are not novel as such: combining existing architectures for the classification and detection tasks, and presenting a new color quantisation network built using attention blocks and a ResNext component. The paper does not go into detail on how the proposed technique differs from the ColorCNN baseline model to which comparison is made. What shortcomings does the prior work have, and how does the current work address these?


# Reproducibility
the authors say code will be released upon publication. Standard datasets are used.
Classification experiments are run on a single GPU, for detection models a setup with 4 GPUs is used.
The replication of the results for the existing methods is less clear: the paper does not discuss how these were obtained.
It is not clear how the color index map for Nafaanra in section 4.5 is obtained, and whether it is accessible to other researchers.


# typos
The manuscript contains quite a few typos, here a are a few:
- Page 3 and 7: "asteroid" --> "asterisk"
- Page 6: in Eq 9 L_High is used, which in the line below L_H is used, these should be the same as I understand it.
- Page 7: in paragraph "evaluation metrics" repetition in "...top-1 classification accuracy and top-1 classification accuracy..."


**Strength And Weaknesses:**

Strengths

+ The paper considers a relatively little studied problem, which might interest a (limited) subset of the ICLR audience.

+ Experiments are conducted on two different tasks (image classification and object detection), and using 4 different datasets for the classification task.

+ The proposed colour quantisation method seems more effective that existing alternative techniques.


Weaknesses

- It is not clear how the results for the existing methods were obtained. Were experiments conducted by the authors of the current paper? Was code provided by the authors of the existing methods? Please clarify the process.

- It was not clear why an input-dependent approach is used to determine the color palette, while an input-independent network is used to map the input image to quantisation indices in the Annotation branch. What happens if we remove the "Palette branch"  and just learn a set of centroids in color space?

- Parts of the paper are hard to follow, in particular the sections 3.4 and 4.4 and 4.5.


**Summary Of The Paper:**

This paper considers learning a quantisation of colour space, and the parallels to how humans learn colour names.
A novel deep network architecture for colour quantisation is proposed, which is learned by optimising accuracy on a task of interest (image classification or object detection) when using images with quantised colours as input, but also takes into account several regularisation terms.
Experiments are conducted on 4 image classification datasets (CIFAR-10/100, STL10, and TinyImages-200) and one object detection dataset (MS-COCO). Comparisons are made to an upper bound of non-quantised colours, and 4 existing methods for colour quantisation.
Generally, the proposed method performs well compared to others on image classification when few colour quantisation levels are used (4-16) and worse otherwise. For object detection the proposed method is compared to only three alternative methods, and compares consistently better or comparable to those, yet substantially worse than the upper bound obtained by not applying any color quantisation.


**Summary Of The Review:**

The paper presents a novel color quantisation technique, and experiments that compare it to a number of baselines.
The main concerns that I have with the paper is a lack of clarity in some parts describing the methods and experiments, and the positioning wrt related prior work. Overall I'm not convinced that the current paper would make a significant impact in the ICLR community.

---

> ### Author Response · Authors · 2022-11-19
> **Response to Reviewer SHap (Part 1)**
>
> > Q1.It was not clear why an input-dependent approach is used to determine the color palette, while an input-independent network is used to map the input image to quantisation indices in the Annotation Branch. What happens if we remove the "Palette Branch" and just learn a set of centroids in color space?
> >
> > Q2.The technical approach lacks a clear motivation. Why is it insufficient to learn several centroids in color space, and assign pixels to the nearest color centroid? What is the motivation for the Palette and Annotation Branches of the proposed approach over simpler alternatives?
>
> Our Palette Branch ensures that reference palette queries are sent to interact with the image and creates the colour palette using both machine preference and image perception features, which maintains the colour specificity of each image. Annotation Branch is to annotate each pixel of an input RGB image with a proper quantised colour index.
>
> If we remove Palette Branch and make the CQFormer just learn a set of centroids in RGB colour space, the accuracy is 30.4%, resulting in a severe drop of 14.7%. We add this experiment to the ablation study in Appendix (Sec. 6.1) of our latest revised version. As shown in Fig. 5 of our latest revised version, Col.(c) is the quantised image generated by just learning a set of centroids in colour space, where the blue sky is represented as light yellow. Therefore, it would make all the images have the same colour palette rather than the same amount of colours, resulting in a loss of perceptual similarity.
>
> > Q3(1).The technical tools / architectures used in the work are not novel as such: combining existing architectures for the classification and detection tasks, and presenting a new color quantisation network built using attention blocks and a ResNext component.
>
> Our novelty is the first to reformulate the problem of colour quantisation as a 3D spatial key-point localisation task among the whole RGB colour space. Hence, we borrow these existing successful architectures.
>
> > Q3(2).The paper does not go into detail on how the proposed technique differs from the ColorCNN baseline model to which comparison is made. What shortcomings does the prior work have, and how does the current work address these?
>
> Traditional colour quantisation methods[3,4,5] are perception-centred and generate a new image which is as perceptually similar as possible to the original image.
>
> Hou et al.[1] creatively proposed a task-centred/machine-centred colour quantisation method, ColorCNN, focusing on clustering the pixels for colour quantisation from a machine learning perspective.
>
> Inspired by [1], we integrate the need for both perception and machine to propose the CQFormer, which echoes the evolution of human languages under the pressure of both communication efficiency and colour perception. Based on previous research, we utilise CQFormer to discover the artificial colour naming systems and their evolution. At the same time, we also contribute to colour quantisation.
>
> > Q4.Purity regularisation: it was not clear to me why a max over pixel positions is taken, this seems to ensure that at least one pixel is strongly assigned to each one of the quantisation bins, rather than that all pixels are assigned to just one color. Did the authors consider, for example, summing over the pixels the entropy of the assignment distribution?
>
> Thanks for your careful reading and suggestions. We have reorganised our statement and have revised the "Purity Regularisation" to "Diversity Regularisation" in the latest revised article. Inspired by Hou et al.[1], the "Diversity Regularisation" aims to encourage the CQFormer to choose as many colours as possible by maximising the maximum probability of each channel. Please refer to Eq.14 on Page. 6. Actually, it is the implicit entropy of the assignment distribution, and we consider using explicit entropy for further research.
>
> References:
>
> [1] Hou Y, Zheng L, Gould S. Learning to structure an image with few colors. InProceedings of the IEEE/CVF Conference on Computer Vision and Pattern Recognition 2020 (pp. 10116-10125).
>
> [2] Carion N, Massa F, Synnaeve G, Usunier N, Kirillov A, Zagoruyko S. End-to-end object detection with transformers. InEuropean conference on computer vision 2020 Aug 23 (pp. 213-229). Springer, Cham.
>
> [3] Paul Heckbert. Color image quantization for frame buffer display. ACM Siggraph Computer Graphics, 16(3):297–307, 198
>
> [4] Michael Gervautz and Werner Purgathofer. A simple method for color quantization: Octree quantization. In New trends in computer graphics, pp. 219–231. Springer, 1988
>
> [5] R. W. Floyd and L. Steinberg. An adaptive algorithm for spatial grayscale. Proceedings of the Society for Information Display, 17, 1976

---

> ### Author Response · Authors · 2022-11-19
> **Response to Reviewer SHap (Part 2)**
>
> > Q5.It is not clear how the results for the existing methods were obtained. Were experiments conducted by the authors of the current paper? Was code provided by the authors of the existing methods? Please clarify the process.
>
> We have conducted the comparison methods, and the code is provided by the authors of the existing methods. Specifically, for ColorCNN[1], we also cascade the ColorCNN and the classifier and jointly train this combination without any pre-train models. For the traditional methods[3,4,5], we conduct comparative experiments as described in [1].
>
> > Q6.It was not clear to me whether the image classification and detection models are trained jointly with the color quantisation model, or whether the classification/detection models are pre-trained without color quantisation, and the used as is without further training them when learning the colour quantisation model.
>
> We cascade the CQFormer and the classifier for the classification task and jointly train this combination from scratch. In other words, the classifier needs to be pre-trained before colour quantisation, and we need to jointly train it with the CQFormer when learning the colour quantisation model. We also cascade the CQFormer and the detector for the detection task and jointly train this combination. The difference is that we only add the Imagenet pre-train model to the Resnet-50[6] backbone of Faster-RCNN[7], and the remaining parts do not add any pre-trained model. We also need to jointly train it with the CQFormer when learning the colour quantisation model.
>
> > Q7.The purpose of Section 4.4. is not clear. For example, the first sentence was unclear to me: "Similar to task centered colorCNN [ref], we are also inferior to the traditional method under a large space." The section does not describe any experiment, but rather seems to be an interpretation of the previous experiments. The text itself should be clarified by clearly pointing to the experimental results on which it is based, and integrating it in the corresponding section. In its current state it did not bring me any additional insights.
>
> Thanks for your careful reading and suggestions. We would merge Sec 4.4 (Performance on Larger Colour Space) into the second paragraph in Sec 4.2 (Classification Task Evaluation).
>
>
> References:
>
> [1] Hou Y, Zheng L, Gould S. Learning to structure an image with few colors. InProceedings of the IEEE/CVF Conference on Computer Vision and Pattern Recognition 2020 (pp. 10116-10125).
>
> [2] Carion N, Massa F, Synnaeve G, Usunier N, Kirillov A, Zagoruyko S. End-to-end object detection with transformers. InEuropean conference on computer vision 2020 Aug 23 (pp. 213-229). Springer, Cham.
>
> [3] Paul Heckbert. Color image quantization for frame buffer display. ACM Siggraph Computer Graphics, 16(3):297–307, 198
>
> [4] Michael Gervautz and Werner Purgathofer. A simple method for color quantization: Octree quantization. In New trends in computer graphics, pp. 219–231. Springer, 1988
>
> [5] R. W. Floyd and L. Steinberg. An adaptive algorithm for spatial grayscale. Proceedings of the Society for Information Display, 17, 1976
>
> [6] Kaiming He, Xiangyu Zhang, Shaoqing Ren, and Jian Sun. Deep residual learning for image recognition. In Proceedings of the IEEE conference on computer vision and pattern recognition, pp.770–778, 2016
>
> [7]  Shaoqing Ren, Kaiming He, Ross Girshick, and Jian Sun. Faster r-cnn: Towards real-time object detection with region proposal networks. Advances in neural information processing systems, 28, 2015

---

> ### Author Response · Authors · 2022-11-19
> **Response to Reviewer SHap (Part 3)**
>
> > Q8.When considering image classification, and increasing the number of colour quantisation levels, the proposed method saturates to an accuracy level that is lower than that for some of the other methods to which is compared (OCTree, MedianCut, MedianCut+dither). A discussion to interpret this result is missing. Is this an inherent limitation of the proposed method? It is due to what? Can it be overcome?
>
> Thanks for your careful reading and suggestions. The CQFormer artificially discovers the colour naming system by considering the needs of both perception and machine via colour quantisation. Therefore, similar to task-centred ColorCNN[1], we are also inferior to the traditional method[3,4,5] under a large space, which is an inherent limitation of the proposed method.  As extensively discussed in [1], this is very normal since traditional methods clusters all pixels to enforce intra-cluster similarity to make a collective decision, while ColorCNN does not formulate the colour quantisation as a clustering problem[1]. For example, the accuracy of quantised images only slightly falls behind original images in a 6-bit colour space.
>
> We argue that it reveals a deeper phenomenon: there is a natural bottleneck (upper bound) in the way humans use language to convey information. As written in Paragraph 2 in Sec.4.2, this implies that discovering more colours not only compromises the principle of efficiency, but also goes contrary to the expectation of better perceptual effects. In other words, the performance of the CQFormer on limited colour categories may hint at the optimal outcome restricted by the unique neurological structure of human vision and cognition, which are in turn reflected in a wide array of languages. To overcome this inherent limitation, we consider combining a competent clustering model with our CQFormer to outperform the traditional methods in a large colour space.
>
> Thanks again for your suggestions, and we look forward to exploring this.
>
>
>
> References:
>
> [1] Hou Y, Zheng L, Gould S. Learning to structure an image with few colors. InProceedings of the IEEE/CVF Conference on Computer Vision and Pattern Recognition 2020 (pp. 10116-10125).
>
> [2] Carion N, Massa F, Synnaeve G, Usunier N, Kirillov A, Zagoruyko S. End-to-end object detection with transformers. InEuropean conference on computer vision 2020 Aug 23 (pp. 213-229). Springer, Cham.
>
> [3] Paul Heckbert. Color image quantization for frame buffer display. ACM Siggraph Computer Graphics, 16(3):297–307, 198
>
> [4] Michael Gervautz and Werner Purgathofer. A simple method for color quantization: Octree quantization. In New trends in computer graphics, pp. 219–231. Springer, 1988
>
> [5] R. W. Floyd and L. Steinberg. An adaptive algorithm for spatial grayscale. Proceedings of the Society for Information Display, 17, 1976
>
> [6] Kaiming He, Xiangyu Zhang, Shaoqing Ren, and Jian Sun. Deep residual learning for image recognition. In Proceedings of the IEEE conference on computer vision and pattern recognition, pp.770–778, 2016
>
> [7]  Shaoqing Ren, Kaiming He, Ross Girshick, and Jian Sun. Faster r-cnn: Towards real-time object detection with region proposal networks. Advances in neural information processing systems, 28, 2015

---

> ### Author Response · Authors · 2022-11-19
> **Response to Reviewer SHap (Part 4)**
>
> > Q9.Section 3.4 the paragraph "colour probability map embedding" was not clear. Is there any fine-tuning of the model involved here, or just an adaptation of an already trained model? I would suggest a more formal description of the method to clarify the exposition.
> >
> > Q10.Similarly the paragraph "Colour evolution" in the same section was unclear to me. Please add a more formal description of the approach here.
>
> We revised our manuscript and added many details, especially the detailed structure in Sec 3.4. The colour probability map embedding is one of our significant innovations. Humans already have different colour systems, so we choose a language that has been carefully researched and take the 1978 Nafaanra three-colour system[2] as an example. To force the CQFormer to match the colour distribution of Nafaanra, we embed the 1978 Nafaanra three-colour system into the latent representation of the CQFormer. Finally, our CQFormer successfully inherits the colour naming system of human language and outputs a similar WCS colour probability map to that of Nafaanra. In other words, we take the 1978 Nafaanra three-colour system as a teacher model and the untrained CQFormer as a student model. Hence, in this knowledge distillation way, we successfully embed the human language into our CQFormer using an additional cross entropy loss in Eq. 17 in our latest revised version.
>
> After embedding the 1978 Nafaanra three-colour system into the latent representation of the CQFormer,  we remove the additional cross-entropy loss. We only utilise the unsupervised signal "Diversity Regularisation”[1] ("Purity Regularisation" in our previous version) to encourage the CQFormer to evolve fine-grained new colour on top of the embedded model. Finally, our method automatically evolves the fourth colour close to yellow-green, matching the basic colour terms theory summarised in different languages[3].
>
> > Q11.The same goes for the last paragraph in section 3.4 "Central colour embedding". It was not clear to me whether this is an alternative regularisation method used during training, or a post-hoc adaptation of the trained model.
>
> We design two embedding solutions to either distil full colour probability map embedding or only representative colours to CQFormer. Alternatively, we could distil less information of the human colour naming system using the central colour embedding. Here, we only embed representative colours and their central colour H and V coordinates. We use it to generate the three WCS colour probability maps in Fig. 1(b). Similar to [1], we also study a scientific problem: does artificial intelligence share the same perceptual mechanism for colours as human beings? Hence, central colour embedding is one of our initial explorations, and we look forward to exploring it further. Thanks for your questions to improve our article.
>
>
> References:
>
> [1] Hou Y, Zheng L, Gould S. Learning to structure an image with few colors. In Proceedings of the IEEE/CVF Conference on Computer Vision and Pattern Recognition 2020 (pp. 10116-10125).
>
> [2] Zaslavsky N, et al. The evolution of color naming reflects pressure for efficiency: Evidence from the recent past. Journal of Language Evolution, 2022. ISSN 2058-458X. doi: 10.1093/jole/lzac
>
> [3] B. Berlin and P. Kay. Basic Color Terms: Their Universality and Evolution. University of California Press, 1969. ISBN 978052007635
>
> [4] Paul Kay, Brent Berlin, Luisa Maffi, William R Merrifield, and Richard Cook. The world color survey. CSLI Publications Stanford, CA, 20
>
> [5] Zaslavsky N, Kemp C, Regier T, Tishby N. Efficient compression in color naming and its evolution. Proceedings of the National Academy of Sciences. 2018 Jul 31;115(31):7937-42.
>
> [6] Paul Heckbert. Color image quantization for frame buffer display. ACM Siggraph Computer Graphics, 16(3):297–307, 198
>
> [7] Michael Gervautz and Werner Purgathofer. A simple method for color quantization: Octree quantization. In New trends in computer graphics, pp. 219–231. Springer, 1988
>
> [8] R. W. Floyd and L. Steinberg. An adaptive algorithm for spatial grayscale. Proceedings of the Society for Information Display, 17, 1976

---

> ### Author Response · Authors · 2022-11-19
> **Response to Reviewer SHap (Part 5)**
>
> > Q12.The colour evolution experiment in section 4.5 it not clearly described. It seem that in the first stage the proposed colour quantisation model is trained to align with the Nafaanra color naming in three colours, and to perform well on the CIFAR-10 classification task. The second stage learns a fourth colour bin, starting from the three colour quantisation. But it seems the design of the experiment already dictates which of the three initial colour bins will be split in the second stage, which seems to strongly direct the outcome of the second training stage. Moreover, it is not clear if the alternative color quantisation methods would exhibit similar behaviour, or whether this is unique to the proposed method.
>
> Thanks for your careful reading and questions. We add experiments to change the order of colour and force the fourth colour to be split from light ('fiNge'), dark ('wOO'), and warm or red-like ('nyiE'), respectively.
>
> During the second stage, the CQFormer automatically evolves the fourth colour that is split from dark ('wOO') and close to yellow-green, matching the basic colour terms theory[3]. However, we are not able to see the fourth colour split from either light ('fiNge') or warm/red-like ('nyiE') in the WCS colour probability map, since only 3.7\% (if split from light ('fiNge')) and 5.9\% (if split from warm/red-like ('nyiE')) of all pixels are assigned to the fourth colour, compared with 23.5\% (if split from dark ('wOO')). Very interestingly, this phenomenon echoes the evolution of the information bottleneck (IB) colour naming systems[5], where the fourth colour should be spilt from dark in the "dark-light-red" colour scheme.
>
> To our knowledge, this colour evolution experiment is unique to the proposed method. For example, the traditional colour quantisation methods[6,7,8] cannot embed colour perceptual knowledge of a certain language into the latent representation.
>
> > Q13.It is not clear how the color index map for Nafaanra in section 4.5 is obtained, and whether it is accessible to other researchers.
>
> The color index map for Nafaanra is collected by World Color Servey (WCS)[4], which  is a collective effort to explore "the existence of a partially fixed evolutionary progression according to which languages gain color terms over time." As introduced in Sec.2, participants were asked to name the colour of each chip to record the colour naming system. The WCS data are available at http://www.icsi.berkeley.edu/wcs/data.html,  and the color index map for Nafaanra is available at https://dx.doi.org/10.7297/X2VH5MCR.[2,5]
>
>
> References:
>
> [1] Hou Y, Zheng L, Gould S. Learning to structure an image with few colors. In Proceedings of the IEEE/CVF Conference on Computer Vision and Pattern Recognition 2020 (pp. 10116-10125).
>
> [2] Zaslavsky N, et al. The evolution of color naming reflects pressure for efficiency: Evidence from the recent past. Journal of Language Evolution, 2022. ISSN 2058-458X. doi: 10.1093/jole/lzac
>
> [3] B. Berlin and P. Kay. Basic Color Terms: Their Universality and Evolution. University of California Press, 1969. ISBN 978052007635
>
> [4] Paul Kay, Brent Berlin, Luisa Maffi, William R Merrifield, and Richard Cook. The world color survey. CSLI Publications Stanford, CA, 20
>
> [5] Zaslavsky N, Kemp C, Regier T, Tishby N. Efficient compression in color naming and its evolution. Proceedings of the National Academy of Sciences. 2018 Jul 31;115(31):7937-42.
>
> [6] Paul Heckbert. Color image quantization for frame buffer display. ACM Siggraph Computer Graphics, 16(3):297–307, 198
>
> [7] Michael Gervautz and Werner Purgathofer. A simple method for color quantization: Octree quantization. In New trends in computer graphics, pp. 219–231. Springer, 1988
>
> [8] R. W. Floyd and L. Steinberg. An adaptive algorithm for spatial grayscale. Proceedings of the Society for Information Display, 17, 1976

---

> ### Author Response · Authors · 2022-11-19
> **Response to Reviewer SHap (Part 6)**
>
> > Q14.The paper refers to studies of color naming in the Nafaanra language, but does not explain why these studies are of particular interest as compared to color naming in other languages, nor how the findings for Nafaanra differ from those for other languages. When referring to analysis of Nafaara in the first paragraph of page 2, or elsewhere, there is no reference to the corresponding study.
>
> Colour naming data for Nafaanra[2] were initially collected in 1978 in Banda Ahenkro, as part of the WCS[3], which is a 3-term system, with terms for light (‘fiNge’), dark (‘wOO’), and warm or red-like (‘nyiE’). In 2018, 40 years after the original WCS data collection, Nafaanra colour naming data were collected again in the same town. As technology and lifestyle changed in the community, the Nafaanra colour naming system changed substantially between 1978 and 2018, becoming more semantically fine-grained by adding new colour terms and adjusting the extension of previously existing terms[2]. Therefore, the Nafaanra language is an excellent example for comparing  the evolution trajectory of human language using the machine.
>
> References:
>
> [1] Hou Y, Zheng L, Gould S. Learning to structure an image with few colors. In Proceedings of the IEEE/CVF Conference on Computer Vision and Pattern Recognition 2020 (pp. 10116-10125).
>
> [2] Zaslavsky N, et al. The evolution of color naming reflects pressure for efficiency: Evidence from the recent past. Journal of Language Evolution, 2022. ISSN 2058-458X. doi: 10.1093/jole/lzac
>
> [3] B. Berlin and P. Kay. Basic Color Terms: Their Universality and Evolution. University of California Press, 1969. ISBN 978052007635
>
> [4] Paul Kay, Brent Berlin, Luisa Maffi, William R Merrifield, and Richard Cook. The world color survey. CSLI Publications Stanford, CA, 20
>
> [5] Zaslavsky N, Kemp C, Regier T, Tishby N. Efficient compression in color naming and its evolution. Proceedings of the National Academy of Sciences. 2018 Jul 31;115(31):7937-42.
>
> [6] Paul Heckbert. Color image quantization for frame buffer display. ACM Siggraph Computer Graphics, 16(3):297–307, 198
>
> [7] Michael Gervautz and Werner Purgathofer. A simple method for color quantization: Octree quantization. In New trends in computer graphics, pp. 219–231. Springer, 1988
>
> [8] R. W. Floyd and L. Steinberg. An adaptive algorithm for spatial grayscale. Proceedings of the Society for Information Display, 17, 1976

---

### Official Review · Reviewer_bh8b · 2022-10-23

**Confidence:** 4
**Correctness:** 3
**Technical Novelty And Significance:** 3
**Empirical Novelty And Significance:** 3
**Recommendation:** 5

**Clarity, Quality, Novelty And Reproducibility:**

- The paper is mostly clear, although some parts can be improved, for example, the paragraph after equation 2.

- As explained in the previous comment, there is some novelty in combining both perception and machine learning techniques for color quantization, but the results are not enough good in my opinion.

- There are some typos here and there, for example:
               º Missing citation to the papers in section 2.1.
               º would generates (page 5)
               º we locates (page 5)

- Authors comment on the realising of code after acceptance, so it should be reproducible.

**Strength And Weaknesses:**

The paper has a positive side, on the study of how can we quantize an image without losing a large accuracy in downstream tasks. This said, this is not a novel research line, as it was already presented in ColorCNN. The main novelty of this work is therefore, trying to include perception into this task.

In terms of results (Figure 5), the proposed method seems to beat ColorCNN is all the cases, and other perception based approaches in low-bin level (1-4). This is positive, but I believe the for number of bins in which there exist an advantage, the accuracy is still not acceptable for the downstream tasks.

Regarding the analysis on the order of appearances of the color names, I believe that the analysis is not of real help; as I am quite convinced the authors are forcing that order of appearance due to the enforcement of the perceptually similarity regularisation. That loss guides the net to actually predict exactly the order of colours in thw WCS, as the probabilities in WCS exactly represent that order of appearance.

**Summary Of The Paper:**

The authors propose a color quantization transformer with the goal of reduding the number of bits needed to allocate the color information, but, at the same time, trying to look at the question: Do machine learning methods make evolve the different color terms as human civilizations?

**Summary Of The Review:**

The paper has interest and that idea might be promising; however, I believe the results of this approach are still not yet comprisng with pure perception approaches, and also, the analysis of the appearance of the color names is perfectly guided by one of the losses applied in the experiment, and therefore cannot be really explained out of the quantization effect.

---

> ### Author Response · Authors · 2022-11-19
> **Response to Reviewer bh8b**
>
> > Q1 In terms of results (Figure 5), the proposed method seems to beat ColorCNN is all the cases, and other perception based approaches in low-bin level (1-4). This is positive, but I believe the for number of bins in which there exist an advantage, the accuracy is still not acceptable for the downstream tasks.
>
> Thanks for your careful reading and suggestions. The CQFormer artificially discovers the colour naming system by considering the needs of both perception and machine via colour quantisation. Therefore, similar to task-centred ColorCNN[1], we are also inferior to the traditional method[3,4,5] under a large space, which is an inherent limitation of the proposed method.  As extensively discussed in [1], this is very normal since traditional methods clusters all pixels to enforce intra-cluster similarity to make a collective decision, while ColorCNN does not formulate the colour quantisation as a clustering problem[1]. For example, the accuracy of quantised images only slightly falls behind original images in a 6-bit colour space.
>
> We argue that it reveals a deeper phenomenon: there is a natural bottleneck (upper bound) in the way humans use language to convey information. As written in Paragraph 2 in Sec.4.2, this implies that discovering more colours not only compromises the principle of efficiency but also goes contrary to the expectation of better perceptual effects. In other words, the performance of the CQFormer on limited colour categories may hint at the optimal outcome restricted by the unique neurological structure of human vision and cognition, which are in turn reflected in a wide array of languages.
>
> Thanks again for your suggestions and we look forward to exploring this.
>
> > Q2 The paper has interest and that idea might be promising; however, I believe the results of this approach are still not yet comprisng with pure perception approaches.
>
> Does the word "comprisng" means "comparing"? In this case, similar to [1], we have compared it with the pure perception approaches in Fig.4.
>
>
>
> > Q3 Regarding the analysis on the order of appearances of the color names, I believe that the analysis is not of real help; as I am quite convinced the authors are forcing that order of appearance due to the enforcement of the perceptually similarity regularisation. That loss guides the net to actually predict exactly the order of colours in thw WCS, as the probabilities in WCS exactly represent that order of appearance.
> >
> > Q4 The analysis of the appearance of the color names is perfectly guided by one of the losses applied in the experiment, and therefore cannot be really explained out of the quantization effect.
>
> The Perceptual Structure Loss is defined in Sec.3.3 as an unsupervised signal. It only guarantees that each item of the colour palette lies in the centre of each cluster, encourages the CQFormer to select more colours, and ensures the pixels in the same cluster are perceptually similar in colour. It does not provide any prior knowledge of the human language, and cannot directly direct the order of colours in the WCS.
>
> **References:**
>
> [1] Hou Y, Zheng L, Gould S. Learning to structure an image with few colors. InProceedings of the IEEE/CVF Conference on Computer Vision and Pattern Recognition 2020 (pp. 10116-10125).
>
> [2] Zaslavsky N, et al. The evolution of color naming reflects pressure for efficiency: Evidence from the recent past. Journal of Language Evolution, 2022. ISSN 2058-458X. doi: 10.1093/jole/lzac
>
> [3] B. Berlin and P. Kay. Basic Color Terms: Their Universality and Evolution. University of California Press, 1969. ISBN 978052007635
>
> [4] Paul Kay, Brent Berlin, Luisa Maffi, William R Merrifield, and Richard Cook. The world color survey. CSLI Publications Stanford, CA, 20
>
> [5] Zaslavsky N, Kemp C, Regier T, Tishby N. Efficient compression in color naming and its evolution. Proceedings of the National Academy of Sciences. 2018 Jul 31;115(31):7937-42.

---

### Official Review · Reviewer_4LP2 · 2022-10-24

**Confidence:** 4
**Correctness:** 4
**Technical Novelty And Significance:** 3
**Empirical Novelty And Significance:** 3
**Recommendation:** 6

**Clarity, Quality, Novelty And Reproducibility:**

This paper is well-written generally but with some minor typo and less clear in some parts. The quality and novelty meet the requirement of the conference. And the method is able to be reproduced.

**Strength And Weaknesses:**

Strength:
(1)	The paper is well-written and description of the proposed method is clear and easy to follow.
(2)	The core problem is novel and has potentially practical usage in vision field.
(3)	The proposed modules are reasonable and the whole experiments indeed support the effectiveness of the method, which shows strong performance improvements compared with previous methods.

Weaknesses:
(1)	From Table 3, we can see that the CQFormer indeed shows superiority under extremely low bits, such as 1-bit and 2-bit, and the performance maintains relatively stable when the color bit decreases. But it fails to compete with traditional methods like MedianCut and OCTree when the color bit is large, such as 5-bit and 6-bit. Could you provide explanation for this phenomenon, and if there are any improvements that can help CQFormer keep consistent performance superiority across all color bits?
(2)	For the experiments part, the value of C is missing in the Training Settings part.
(3)	The description for the Palette Branch is not enough. For example, the detailed structure of Cross Attention and FFN is not specified or formulated. And I do not know if there is a shortcut alongside or a normalization along with Cross Attention and FFN.
(4)	Some typo such as “to to” in the 3rd paragraph of the Introduction, “a explicit” in the last but one paragraph of Page 4. And the author should leave some blank between Figure 5 and the following half paragraph.
(5)	I am doubt with the name “transformer” because it seems that the author only implement one “Cross Attention + FFN” layer, which is only part of Decoder of standard Transformer. In my opinion, only the stacks of multiple Encoder layers or Decoder layers could be named as “Transformer”. The implementation in this paper seems like only adopting Cross Attention Mechanism. So I suggest change the name to the other one without “Transformer”.


**Summary Of The Paper:**

This paper proposes an end-to-end color quantization transformer that is able to discover the color naming system under the need of perception and machine. The framework contains two branches, respectively for structure mining and color localization. The proposed CQFormer quantizes the color space of the input image and the generated quantized image can maintain recognition accuracy, which is of practical application such as images compression. Besides, the proposed method can provide consistent evolution pattern between the color system and the basic color terms across human languages. The experiments also demonstrate the effectiveness of the method.

**Summary Of The Review:**

This paper is generally of good quality and the novel method is supported by the sufficient experiments. I point out some weaknesses of the paper and ask for paper revision. In conclusion, this paper meets the requirement of the conference and I vote for Accept.

---

> ### Author Response · Authors · 2022-11-19
> **Response to Reviewer 4LP2 (Part 1)**
>
> > Q1 From Table 3, we can see that the CQFormer indeed shows superiority under extremely low bits, such as 1-bit and 2-bit, and the performance maintains relatively stable when the color bit decreases. But it fails to compete with traditional methods like MedianCut and OCTree when the color bit is large, such as 5-bit and 6-bit. Could you provide explanation for this phenomenon, and if there are any improvements that can help CQFormer keep consistent performance superiority across all color bits?
>
> Thanks for your careful reading and suggestions. The CQFormer artificially discovers the colour naming system by considering the needs of both perception and machine via colour quantisation. Therefore, similar to task-centred ColorCNN[1], we are also inferior to the traditional method[3,4,5] under a large space, which is an inherent limitation of the proposed method.  As extensively discussed in [1], this is very normal since traditional methods clusters all pixels to enforce intra-cluster similarity to make a collective decision, while ColorCNN does not formulate the colour quantisation as a clustering problem[1]. For example, the accuracy of quantised images only slightly falls behind original images in a 6-bit colour space.
>
> We argue that it reveals a deeper phenomenon: there is a natural bottleneck (upper bound) in the way humans use language to convey information. As written in Paragraph 2 in Sec.4.2, this implies that discovering more colours not only compromises the principle of efficiency but also goes contrary to the expectation of better perceptual effects. In other words, the performance of the CQFormer on limited colour categories may hint at the optimal outcome restricted by the unique neurological structure of human vision and cognition, which are in turn reflected in a wide array of languages. For improvement, we consider combining a competent clustering model with our CQFormer to outperform the traditional methods in a large colour space.
>
> Thanks again for your suggestions and we look forward to exploring this.
>
> > Q2 For the experiments part, the value of C is missing in the Training Settings part.
>
> In the experiments, the colour bit ranging from 1 to 6 is the exponential power to the base 2 of C. Therefore,  the value of C  is 2, 4, 8, 16, 32, and 64, respectively. We have added this to the implement details of Sec. 4.1 in our latest revised version.
>
> > Q3 The description for Palette Branch is not enough. For example, the detailed structure of Cross Attention and FFN is not specified or formulated. And I do not know if there is a shortcut alongside or a normalization along with Cross Attention and FFN.
>
> We revise our manuscript and add lots of details. Please refer to Sec 3.2 in our latest revised version.
>
> > Q4 Some typo such as “to to” in the 3rd paragraph of the Introduction, “a explicit” in the last but one paragraph of Page 4. And the author should leave some blank between Figure 5 and the following half paragraph.
>
> Thanks for your careful reading and suggestions. We have removed the typos in our latest revised manuscript.
>
>
> **References:**
>
> [1] Hou Y, Zheng L, Gould S. Learning to structure an image with few colors. InProceedings of the IEEE/CVF Conference on Computer Vision and Pattern Recognition 2020 (pp. 10116-10125).
>
> [2] Carion N, Massa F, Synnaeve G, Usunier N, Kirillov A, Zagoruyko S. End-to-end object detection with transformers. In European conference on computer vision 2020 Aug 23 (pp. 213-229). Springer, Cham.
>
> [3] Paul Heckbert. Color image quantization for frame buffer display. ACM Siggraph Computer Graphics, 16(3):297–307, 198
>
> [4] Michael Gervautz and Werner Purgathofer. A simple method for color quantization: Octree quantization. In New trends in computer graphics, pp. 219–231. Springer, 1988
>
> [5] R. W. Floyd and L. Steinberg. An adaptive algorithm for spatial grayscale. Proceedings of the Society for Information Display, 17, 1976
>
> [6] Kaiming He, Xiangyu Zhang, Shaoqing Ren, and Jian Sun. Deep residual learning for image recognition. In Proceedings of the IEEE conference on computer vision and pattern recognition, pp.770–778, 2016
>
> [7]  Shaoqing Ren, Kaiming He, Ross Girshick, and Jian Sun. Faster r-cnn: Towards real-time object detection with region proposal networks. Advances in neural information processing systems, 28, 2015
>
> [8] Sun P, Zhang R, Jiang Y, Kong T, Xu C, Zhan W, Tomizuka M, Li L, Yuan Z, Wang C, Luo P. Sparse r-cnn: End-to-end object detection with learnable proposals. InProceedings of the IEEE/CVF conference on computer vision and pattern recognition 2021 (pp. 14454-14463).

---

> ### Author Response · Authors · 2022-11-19
> **Response to Reviewer 4LP2 (Part 2)**
>
> > Q5 I am doubt with the name “transformer” because it seems that the author only implement one “Cross Attention + FFN” layer, which is only part of Decoder of standard Transformer. In my opinion, only the stacks of multiple Encoder layers or Decoder layers could be named as “Transformer”. The implementation in this paper seems like only adopting Cross Attention Mechanism. So I suggest change the name to the other one without “Transformer”.
>
> We use the name “transformer” inspired by DEtection TRansformer (DETR)[2], and we are the first to reformulate the problem of colour quantisation as a 3D spatial key-point localisation task among the whole RGB colour space. Using the cross attention mechanism, our “Cross Attention + FFN” layer can also be stacked in multiple layers. However, we only use one layer to meet the demand for lighter weight.
>
> **References:**
>
> [1] Hou Y, Zheng L, Gould S. Learning to structure an image with few colors. InProceedings of the IEEE/CVF Conference on Computer Vision and Pattern Recognition 2020 (pp. 10116-10125).
>
> [2] Carion N, Massa F, Synnaeve G, Usunier N, Kirillov A, Zagoruyko S. End-to-end object detection with transformers. In European conference on computer vision 2020 Aug 23 (pp. 213-229). Springer, Cham.
>
> [3] Paul Heckbert. Color image quantization for frame buffer display. ACM Siggraph Computer Graphics, 16(3):297–307, 198
>
> [4] Michael Gervautz and Werner Purgathofer. A simple method for color quantization: Octree quantization. In New trends in computer graphics, pp. 219–231. Springer, 1988
>
> [5] R. W. Floyd and L. Steinberg. An adaptive algorithm for spatial grayscale. Proceedings of the Society for Information Display, 17, 1976
>
> [6] Kaiming He, Xiangyu Zhang, Shaoqing Ren, and Jian Sun. Deep residual learning for image recognition. In Proceedings of the IEEE conference on computer vision and pattern recognition, pp.770–778, 2016
>
> [7]  Shaoqing Ren, Kaiming He, Ross Girshick, and Jian Sun. Faster r-cnn: Towards real-time object detection with region proposal networks. Advances in neural information processing systems, 28, 2015
>
> [8] Sun P, Zhang R, Jiang Y, Kong T, Xu C, Zhan W, Tomizuka M, Li L, Yuan Z, Wang C, Luo P. Sparse r-cnn: End-to-end object detection with learnable proposals. InProceedings of the IEEE/CVF conference on computer vision and pattern recognition 2021 (pp. 14454-14463).

---

### Official Review · Reviewer_hYs4 · 2022-10-25

**Confidence:** 2
**Correctness:** 2
**Technical Novelty And Significance:** 3
**Empirical Novelty And Significance:** 2
**Recommendation:** 6

**Clarity, Quality, Novelty And Reproducibility:**

Novelty: The ideas presented in this paper (i.e. CQFormer, loss formulation, and regularizers) are original to my knowledge. I found Perceptually Similarity Regularisation intuitive and useful.

Reproducibility: I did not check for reproducibility very carefully. There are some technical details that need clarity. The datasets used are publicly available. The paper promises to release the code.

Clarity: The paper is not well-motivated. What is the impact of solving this task? Why someone would work on this problem? The motivation should be such that a reader from an outside area is able to appreciate the work. In the current write-up, it is highly unclear. Even the abstract does not make clear sense. The same applies to the title. It is much later possibly after going through the other related works such as colorCNN and the experiment section the motivation became somewhat clear.


**Strength And Weaknesses:**

Strength:
+ The ideas presented in this paper (i.e. CQFormer, loss formulation, and regularizers) are original to my knowledge. I found Perceptually Similarity Regularisation intuitive and useful.

+ The experimental results show the effectiveness of the proposed approach for both detection as well as classification task. It is impressive that the proposed quantization with just a few bits is able to achieve a decent AP on MS-COCO object detection. The performance is clearly better than the related publications.

Weakness:
- Upper bounds are misleading. For example, why MS-COCO detection upper bound is this low? Is it the SOTA object detection? I do not think so. If it is the upper bound for the method then it has to be stated and compared against SOTA.

- Are the weights of regularization terms (R) empirically decided? How sensitive is the model for a different alpha, beta, and gamma? Why alpha is chosen as 0 for the detection task?

- Terms in equation 7 are not clearly defined. Such as what is c, (u,v)? Without these definitions, it is difficult to assess the technical correctness of this equation.

- How sensitive is the model to different noises such as color jitter or Gaussian blur? Such studies are important for assessing the robustness of the proposed approach.



**Summary Of The Paper:**

The paper presents CQFormer: a color quantization transformer. The goal is to artificially discover the color naming system so that even with less bit representation acceptable performance on core vision tasks such as classification and detection can be achieved. Experiments show that the proposed method significantly outperforms related approaches for detection and classification tasks on public benchmarks.



**Summary Of The Review:**

Though the method has some originality and achieves superior performance to related work, there are several weaknesses such as limited ablations, experimental choices, missing technical details, and poor presentation of the paper.

---

> ### Author Response · Authors · 2022-11-19
> **Response to Reviewer hYs4 (Part 1)**
>
> > Q1. Upper bounds are misleading. For example, why MS-COCO detection upper bound is this low? Is it the SOTA object detection? I do not think so. If it is the upper bound for the method then it has to be stated and compared against SOTA.
>
> In order to verify the general human language model, we use the general mainstream object detection model, Faster-RCNN[7]. We also utilise the sparse-rcnn[8] detector as the upper bound of detection, and the results are shown in Sec.6.2 and the table bellow.
>
> | Method              |  1-bit   |  2-bit   |  3-bit   |  4-bit   |  5-bit   |  6-bit   | full colour(24-bit) |
> | :------------------ | :------: | :------: | :------: | :------: | :------: | :------: | :-----------------: |
> | Upper bound         |    -     |    -     |    -     |    -     |    -     |    -     |        45.0         |
> | MedianCut[3]        |   11.5   |   12.7   |   15.4   |   17.0   |   20.4   |   23.2   |          -          |
> | MedianCut+Dither[5] |   12.3   |   13.8   |   15.2   |   19.6   |   21.8   |   25.6   |          -          |
> | OCTree[4]           |   10.7   |   13.4   |   13.2   |   16.7   |   18.9   |   22.8   |          -          |
> | **CQFormer**        | **13.9** | **16.5** | **18.8** | **21.5** | **27.5** | **29.8** |          -          |
>
> > Q2. About the settings details of the Regulation term R's weights Are the weights of regularization terms (R) empirically decided? How sensitive is the model for a different alpha, beta, and gamma? Why alpha is chosen as 0 for the detection task?
>
> The weights of regularisation terms are decided through plenty of experiments. Hence the alpha is chosen as 0 for better detection performance. Different values of them are not sensitive since removing one of them has a limited impact. However, a considerable accuracy drop of -7.8% occurred when we removed them. We perform a series of ablation studies, and please refer to Sec. 6.1.
>
> > Q3. About the Eq.7 Terms in equation 7 are not clearly defined. Such as what is c, (u,v)? Without these definitions, it is difficult to assess the technical correctness of this equation.
>
> Thanks for your careful reading and suggestions. We have reorganised our statement and have revised the "Purity Regularisation" to "Diversity Regularisation" in the latest revised article. Inspired by Hou et al.[1], "Diversity Regularisation" aims to encourage the CQFormer to choose as many colours as possible by maximising the maximum probability of each channel. Please refer to the "Diversity Regularisation" of Sec.3.3 in our latest revised version.
>
>
> **References:**
>
> [1] Hou Y, Zheng L, Gould S. Learning to structure an image with few colors. InProceedings of the IEEE/CVF Conference on Computer Vision and Pattern Recognition 2020 (pp. 10116-10125).
>
> [2] Carion N, Massa F, Synnaeve G, Usunier N, Kirillov A, Zagoruyko S. End-to-end object detection with transformers. In European conference on computer vision 2020 Aug 23 (pp. 213-229). Springer, Cham.
>
> [3] Paul Heckbert. Color image quantization for frame buffer display. ACM Siggraph Computer Graphics, 16(3):297–307, 198
>
> [4] Michael Gervautz and Werner Purgathofer. A simple method for color quantization: Octree quantization. In New trends in computer graphics, pp. 219–231. Springer, 1988
>
> [5] R. W. Floyd and L. Steinberg. An adaptive algorithm for spatial grayscale. Proceedings of the Society for Information Display, 17, 1976
>
> [6] Kaiming He, Xiangyu Zhang, Shaoqing Ren, and Jian Sun. Deep residual learning for image recognition. In Proceedings of the IEEE conference on computer vision and pattern recognition, pp.770–778, 2016
>
> [7]  Shaoqing Ren, Kaiming He, Ross Girshick, and Jian Sun. Faster r-cnn: Towards real-time object detection with region proposal networks. Advances in neural information processing systems, 28, 2015
>
> [8] Sun P, Zhang R, Jiang Y, Kong T, Xu C, Zhan W, Tomizuka M, Li L, Yuan Z, Wang C, Luo P. Sparse r-cnn: End-to-end object detection with learnable proposals. InProceedings of the IEEE/CVF conference on computer vision and pattern recognition 2021 (pp. 14454-14463).
>
> [9] Federico Camposeco, Andrea Cohen, Marc Pollefeys, and Torsten Sattler. Hybrid scene compression for visual localization. In Proceedings of the IEEE/CVF Conference on Computer Vision and Pattern Recognition, pp. 7653–7662, 2019.

---

> ### Author Response · Authors · 2022-11-19
> **Response to Reviewer hYs4 (Part 2)**
>
> > Q4. About the robustness of CQFormerHow sensitive is the model to different noises such as color jitter or Gaussian blur? Such studies are important for assessing the robustness of the proposed approach.
>
> Our CQFormer is insensitive and robust to different noises, such as colour jitter and Gaussian blur. We add this experiment to the ablation studies of Appendix, and please refer to Sec. 6.1. For example, on the CIFAR10 classification dataset, we add a colour jitter to the colour quantised image and achieve 79.3%, 81.6%, 83.4%, and 84.6% top-1 accuracy from 1-bit to 4-bit colour quantisation. The colour jitter causes a little drop of 1.4%, 1.5%, 0.4%, and 0.6%, respectively, which proves the robustness of CQFormer.
>
> > Q5. About the motivation of this workThe paper is not well-motivated. What is the impact of solving this task? Why someone would work on this problem? The motivation should be such that a reader from an outside area is able to appreciate the work. In the current write-up, it is highly unclear. Even the abstract does not make clear sense. The same applies to the title. It is much later possibly after going through the other related works such as colorCNN and the experiment section the motivation became somewhat clear.
>
> The International Conference on Learning Representations encourages researchers to explore how neural networks learn meaningful and useful latent representations. We select a broad view to directly match the latent represtation of human language knowledge.
>
> Just as the ICLR encourages, we aim to explore whether artificial intelligence could discover and evolve a similar colour-naming system to human language. We discover the colour perception in the neural networks from the latent representation of human languages and force the latent representation of neural networks to match that of human languages.
>
> Besides, our CQFormer is a continuous research of colour quantisation[1,3,4,5,9] and the latest development, which also effectively compresses image storage while maintaining high performance in high-level recognition tasks such as classification and detection.
>
> **References:**
>
> [1] Hou Y, Zheng L, Gould S. Learning to structure an image with few colors. InProceedings of the IEEE/CVF Conference on Computer Vision and Pattern Recognition 2020 (pp. 10116-10125).
>
> [2] Carion N, Massa F, Synnaeve G, Usunier N, Kirillov A, Zagoruyko S. End-to-end object detection with transformers. InEuropean conference on computer vision 2020 Aug 23 (pp. 213-229). Springer, Cham.
>
> [3] Paul Heckbert. Color image quantization for frame buffer display. ACM Siggraph Computer Graphics, 16(3):297–307, 198
>
> [4] Michael Gervautz and Werner Purgathofer. A simple method for color quantization: Octree quantization. In New trends in computer graphics, pp. 219–231. Springer, 1988
>
> [5] R. W. Floyd and L. Steinberg. An adaptive algorithm for spatial grayscale. Proceedings of the Society for Information Display, 17, 1976
>
> [6] Kaiming He, Xiangyu Zhang, Shaoqing Ren, and Jian Sun. Deep residual learning for image recognition. In Proceedings of the IEEE conference on computer vision and pattern recognition, pp.770–778, 2016
>
> [7]  Shaoqing Ren, Kaiming He, Ross Girshick, and Jian Sun. Faster r-cnn: Towards real-time object detection with region proposal networks. Advances in neural information processing systems, 28, 2015
>
> [8] Sun P, Zhang R, Jiang Y, Kong T, Xu C, Zhan W, Tomizuka M, Li L, Yuan Z, Wang C, Luo P. Sparse r-cnn: End-to-end object detection with learnable proposals. InProceedings of the IEEE/CVF conference on computer vision and pattern recognition 2021 (pp. 14454-14463).
>
> [9] Federico Camposeco, Andrea Cohen, Marc Pollefeys, and Torsten Sattler. Hybrid scene compression for visual localization. In Proceedings of the IEEE/CVF Conference on Computer Vision and Pattern Recognition, pp. 7653–7662, 2019.

---

### Decision · Program_Chairs · 2023-01-20

**Decision:**

Reject

**Justification For Why Not Higher Score:**

The AC agrees with the two reviewers on their remaining concerns. Specifically, the color naming/evolution experiment could have been more rigorous: comparisons without there human coloring systems could be conducted, baseline methods should be compared. The perceptual loss uses a particular color space, which determines the color clusters. Would using a different color space lead to different clusters, that are similar to other human coloring systems?  Finally, the numerical results were also not impressive; while the proposed method performs better on low-bit regime (1-3 bits), it performs worse on 4-6 bits, and its performance appears to saturate. Thus using more bits (7-16) may not lead to improved performance, which is worrying. Perhaps some hyperparameters could be tuned to better trade off the color-based and task-based losses.

Overall, the paper needs a major revision to fix these issues.


**Justification For Why Not Lower Score:**

n/a

**Metareview: Summary, Strengths And Weaknesses:**

**Summary**: The authors propose a color quantization transformer with the goal of reducing the number of bits needed to allocate the color information, but, at the same time, trying to look at the question: Do machine learning methods make evolve the different color terms as human civilizations?
The paper initially received mixed reviews 3566.

**Strengths**: The paper considers a relatively little studied problem, which might interest a (limited) subset of the ICLR audience. Experiments are conducted on two different tasks (image classification and object detection), and using 4 different datasets for the classification task. The proposed colour quantisation method seems more effective that existing alternative techniques in some scenarios.

**Weaknesses**:  The major concerns were:

1. upper bounds for MS-COCO detection are misleading. Is the SOTA detector really used? [hYs4]
2. how are the weights for the regularization (R) set? How sensitive is the model to the hyperparams (alpha, beta, gamma)? Why is alpha=0 for detection? [hYs4]
3. How sensitive is the model to color jitter or Gaussian blur? [hYs4]
4. Presentation issues: Terms in equations are not defined well. [hYs4]
5. the paper is not well motivated. [hYs4]
6. Why is the proposed method better than other methods in the low-bit regime, but worse in the high-bit regime? The proposed method saturates to an accuracy level below other methods. [4LP2, SHap]
7. The description of the palette branch needs improvement. [4LP2]
8. Other presentation issues. [4LP2]
9. Although the performance is better than other methods on low-bit regime, this is not a useful regime due to the poor accuracy in the downstream task. [bh8b]
10. the analysis on the order of color names is not helpful, because it is caused by the perceptually similarity regularization. [bh8b]
11. how were the results of existing methods obtained? [SHap]
12. why use an input dependent approach to determine the color palette? [SHap]
13. What if we remove the "palette branch" and just learn the centroids in the color space? [SHap]
14. descriptions are unclear, Sec 3.4, 4.4, 4.5. [SHap]
15. are the image classification and detection models jointly trained with the color quantization model? [SHap]
16. color evolution experiment in Sec 4.5 is not clear. The design of the experiment (3 clusters, then adding a 4th) dictates the outcome. Could other methods exhibit the same behavior? [SHap]
17. no explanation about why the study of color naming in Nafaarnra is of interest. [SHap]
18. limited discussion of related work. [SHap]
19. technical presentation lacks motivation. [SHap]
20. why use max instead of sum for purity regularization? [SHap]
21. the technical methodology is combination of existing works. [SHap]

**Discussion:** The authors wrote a response to address these concerns. During the discussion among reviewers, Reviewers SHap and bh8b were not convinced. Specifically, Reviewer SHap mentioned:
  - One point that stands out for me is that the approach leads to a per-image colour quantisation: that is the same number of colours is used to quantise each image, but the palette can differ per image (as per the palette branch). I asked this in my review, and authors confirmed in the rebuttal. This means that the quantisation does not correspond to a colour naming scheme, as there is not fixed set of colour bins to which is quantised. This puts the complete approach in question for me. (Points 12, 13)
  - On the experimental comparison: the authors say they used code from previous work te compare to those methods, but do not provide links to repositories where such code can be found. (Point 11)
  - I'm not convinced why only a single human color naming system is compared to with only 3 or 4 colour names. A stronger case of similarity to human colour naming could be made by comparing other human colour systems. (Point 17)
  - Also after the rebuttal, it was not clear to me why the colour naming evolution experiment cannot be conducted with baseline methods, so that comparison to those can be made.  (Point 16)
  - Overall, I feel the paper has a number of significant shortcomings, and needs a complete overhaul to address these before being ready for publication.

Reviewer bh8b mentioned:
  - I am not convinced at all for the answer given to me about the perceptual loss not being important in the selection of colours. Basically, by enforcing closer colors in the color space, you are somehow already marking the size and distance among clusters you can have in that space. This is specially worrying as distances are also computed in Munsell space (and not in a perceptual uniform one as CIELAB or CIECAM). (Point 10)
  - Also, I still believe the numerical results are not good enough. (Points 6, 9)




**Summary Of Ac-Reviewer Meeting:**

n/a